# Computational fluid dynamic analysis of the nasal respiratory function before and after postero-superior repositioning of the maxilla

**Misaki Aoyagi**[1], **Marie Oshima**[2]*, **Masamichi Oishi**[2], **Soma Kita**[1], **Koichi Fujita**[1,3], **Haruki Imai**[1,3], **Shuji Oishi**[1], **Hiroko Ohmori**[1], **Takashi Ono**[1]

1 Department of Orthodontic Science, Graduate School of Medical and Dental Sciences, Tokyo Medical and Dental University (TMDU), Tokyo, Japan, 2 Institute of Industrial Science, The University of Tokyo, Tokyo, Japan, 3 Department of Oral and Maxillofacial Surgery, Yokohama City University Medical Center, Kanagawa, Japan

* marie@iis.u-tokyo.ac.jp

## Abstract

Morphological changes in the upper airway and the resulting alteration in the nasal respiratory function after jawbone repositioning during orthognathic surgery have garnered attention recently. In particular, nasopharyngeal stenosis, because of the complex influence of both jaws, the effects of which have not yet been clarified owing to postero-superior repositioning of the maxilla, may significantly impact sleep and respiratory function, necessitating further functional evaluation. This study aimed to perform a functional evaluation of the effects of surgery involving maxillary repositioning, which may result in a larger airway resistance if the stenosis worsens the respiratory function, using CFD for treatment planning. A model was developed from CT images obtained preoperatively (PRE) and postoperatively (POST) in females (n = 3) who underwent maxillary postero-superior repositioning using Mimics and ICEM CFD. Simultaneously, a model of stenosis (STENOSIS) was developed by adjusting the severity of stenosis around the PNS to simulate greater repositioning than that in the POST. Inhalation at rest and atmospheric pressure were simulated in each model using Fluent, whereas pressure drop (ΔP) was evaluated using CFD Post. In this study, ΔP was proportional to airway resistance because the flow rate was constant. Therefore, the magnitude of ΔP was evaluated as the level of airway resistance. The ΔP in the airway was lower in the POST compared to the PRE, indicating that the analysis of the effects of repositioning on nasal ventilation showed that current surgery is appropriate with respect to functionality, as it does not compromise respiratory function. The rate of change in the cross-sectional area of the mass extending pharynx (α) was calculated as the ratio of each neighboring section. The closer the α-value is to 1, the smaller the ΔP, so ideally the airway should be constant. This study identified airway shapes that are favorable from the perspective of fluid dynamics.

**Data Availability Statement:** The underlying data cannot be shared publicly due to ethical restrictions. Data are available upon request from the ethics committee as follows: Dental Research

Ethics Committee of Tokyo Medical and Dental University E-mail: d-hyoka.adm@tmd.ac.jp Url: https://tmdu-berc.jp.

**Funding:** • Names of the authors who received each award: Soma Kita, Shuji Oishi • Grant numbers awarded to each author: 19K19261, 19K21364 • The full name of each funder: Japan Society for the Promotion of Science (JSPS) • URL of funder website: https://www.jsps.go.jp/ The funders had no role in study design, data collection and analysis, decision to publish, or preparation of the manuscript.

**Competing interests:** The authors have declared that no competing interests exist.

## Introduction

The goal of orthognathic surgery involving the resection of the jawbone is to improve masticatory function and achieve stable occlusion when orthodontic treatment alone cannot correct discrepancies related to the dentition, jawbone, or face. This surgery entails repositioning of the jawbone, which is a hard tissue, along with changes in the soft tissues and airway. Postoperative airway stenosis may have a significant impact on sleep and respiratory function [1, 2]. Therefore, the morphological changes in the airway and the resulting alteration in nasal respiratory function have recently attracted the attention of researchers [3, 4].

Morphological evaluation of nasal respiratory function has been performed using nasal tests [5–7] (e.g., rhinomanometry and acoustic rhinometry), roentgenographic cephalometric analysis [1, 8, 9], and computed tomography (CT) [3, 10, 11]. However, functional or morphological evaluation of a specific site of the airway cannot be generalized to that of the ventilation of the entire airway, owing to its morphological characteristics (i.e., long, narrow, and complicated tubular structure). Moreover, the optimal evaluation method remains elusive because airway obstruction due to airway stenosis can occur at any site, including the nasal cavity, nasopharynx, oropharynx, and hypopharynx. The jawbone and the soft tissues and airway that are moved during orthognathic surgery can be associated with all of these source sites because of their location. Mouth breathing during sleep in the presence of nasal respiratory problems reduces the activity of the airway dilator muscles and the diameter of the airway lumen, increasing the risk of obstructive sleep apnea (OSA) [12].

OSA has also been associated with maxillofacial morphology [9, 13, 14]. A high incidence of mandibular skeletal prognathism (Class III) has been reported in Japan; it is treated using posterior repositioning of the mandible (single-jaw mandibular setback osteotomy). When upper and lower jaw osteotomy (two-jaw surgery) is applied because of a severe discrepancy of the jaws, the maxilla is moved upward and/or backward and the mandible is moved backward. Although there is a lack of clear evidence that corrective jaw surgery causes OSA, it is clear that posterior surgical repositioning of the mandible leads to postoperative narrowing of the upper airway. Therefore, except nasopharynx, several studies have reported the relationship between stenosis of the nasal cavity, oropharynx, or hypopharynx and OSA [3, 8, 15, 16]. On the other hand, in the case of maxillary skeletal prognathism (Class II), the maxilla is moved upward and/or backward or the mandible is moved forward in single-jaw surgery. If two-jaw surgery is required, the maxilla is moved upward and/or backward and the mandible is moved forward. One study reported that patients with Class II have smaller pharyngeal airway volume due to the maxillofacial morphology, which is more likely to lead to OSA compared to the Class I and III skeletal relationships [17]. However, the effect of surgical repositioning of the jaw on OSA has not been elucidated. Despite maxillary impaction, anterior repositioning of the mandible in patients with a Class II skeletal relationship may improve the respiratory status during sleep by expanding the volume of the pharynx. On the other hand, in Class II, posterior and/or superior repositioning of the maxilla may lead to narrowing of the nasal cavity and nasopharynx as was observed in Class III with a reduction in the volume of the airway in the nasal cavity and the most posterior point on the posterior nasal spine (PNS) [4]. Nasal airflow and the cross-sectional area of the nasal cavity decrease when the degree of maxillary impaction exceeds a certain limit [2]. Thus, repositioning of the maxilla may reduce the volume of the entire upper airway changes with the degree of maxillary impaction and mandibular position. The nasopharynx is thought to be susceptible to the movement of both jaws due to its location. Therefore, preventing the reduction in overall ventilation of the upper airway necessitates the evaluation of the nasopharynx, on which the effects of stenosis have not yet been clarified.

Therefore, this study focused on corrective jaw surgery involving the postero-superior repositioning of the maxilla, which is accompanied by a high risk of morphological changes in the nasopharyngeal airway. Maxillary prognathism and vertical maxillary excess (VME) without significant mandibular anomalies is an indication for corrective jaw surgery with postero-superior repositioning with maxillary osteotomy alone, without osteotomy of the mandibular ramus. Operative stress arising from osteotomy of the mandibular ramus and repositioning of the distal fragments of the mandible (e.g., the body of the mandible) may lead to postoperative development or exacerbation of progressive condylar resorption in patients with maxillary prognathism, VME, and significant deformation of the condyle [18–20]. Therefore, maxillary osteotomy alone (without mandibular osteotomy) is recommended to prevent relapse of the mandible [21, 22]. The maxilla is repositioned posteriorly with impaction to improve the facial appearance and occlusion, whereas the mandible undergoes reactionary counter-clockwise rotation during postero-superior repositioning, to achieve occlusion with the maxilla (Fig 1). The improvements in the safety of surgical methods owing recent advancements and development of the ultrasonic osteotomy device have facilitated an increase in the degree of postero-superior repositioning of the maxilla [23–25] (as shown in Fig 1, pink area), leading to higher deformation and narrowing of the nasopharyngeal airway and a higher risk of nasopharyngeal stenosis. However, it is difficult to predict the morphological changes in the airway before surgery. Therefore, the aim of this study was to perform a functional evaluation of the effects of corrective jaw surgery involving

## Jawbone movement          Airway changes

**Fig 1. Postero-superior repositioning of the maxilla and mandibular autorotation.** Surgical impaction of the maxilla and the reaction of the mandible and the associated changes in the airway are illustrated schematically. Notes: black line, pre-surgery (before mandibular autorotation); red line, post-surgery; blue circle, center of mandibular autorotation; blue arrow, direction of autorotation; green line, after mandibular autorotation; gray line; post-surgery in the figure of airway changes; pink area, preoperative nasal cavity; pink hatched area, postoperative nasal cavit; pink arrow, direction of nasal cavity change; purple area, preoperative nasopharyngeal airway; purple hatched area, postoperative nasopharyngeal airway; purple arrow, direction of pharyngeal change.

postero-superior repositioning of the maxilla on nasal respiratory function using computational fluid dynamics for the purpose of treatment planning. The new insights acquired in this study may improve understanding of the pathogenesis of OSA and the effect of orthognathic surgery.

## Materials and methods

### Participants

This study was conducted under approval of the Ethics Committee of Tokyo Medical and Dental University (TMDU) (approval number: D2018-003) and the Institutional Ethical Review Board of the School of Medicine, Yokohama City University (approval number: B110512003). All patients provided written informed consent prior to participation.

This study enrolled three patients, all female, diagnosed with maxillary skeletal prognathism, who underwent Le Fort I osteotomy including postero-superior repositioning of the maxilla [the amount of repositioning was measured with the maxillary central incisor (U1) and first maxillary molar (U6) as reference], and complete postoperative orthognathic surgery (age at surgery, 21 to 36 years and body mass index (BMI) 17.8 to 19.4 kg/m$^2$) at the Yokohama City University Medical Center between 2012 and 2015.

The exclusion criteria were as follows: patients who underwent repositioning of the mandible; patients with a history of facial fractures, tumors, cystic lesions, etc.; patients with congenital anomalies or endocrine disease; patients with significant deviation of the jawbone; and patients with significant nasal deviation.

### Three-dimensional models

This study utilized CT images acquired immediately before and 1 year after surgery using the Aquilion 16 scanner (Toshiba Medical Systems, Tokyo, Japan). The slice thickness was set at 1.0 mm, and the slice width and height were $512 \times 512$ pixels. The pixel size was $4.68 \times 10^{-4}$ m. Imaging was performed while the patient was awake. The head was positioned with the Frankfort horizontal (FH) plane horizontal to the floor. Imaging was performed with the teeth in occlusion, while the breath held and the mouth closed as much as possible. The CT imaging data were saved in the Digital Imaging and Communications in Medicine (DICOM) format.

Segmentation of the upper airways was performed on the basis of the Hounsfield unit, a measure of the electron density of the tissue, assigned to each pixel of the saved DICOM images imported into Mimics (Materialise, Leuven, Belgium) to generate a three-dimensional (3D) model. The threshold was adjusted to obtain a clear image of the airway after eliminating the imaging artifacts. The 3D model was generated in the area between the nasal aperture and subglottis, except for the paranasal sinus, and a "driver" was added to reduce the effect of the inlet and outlet boundary conditions (Fig 2).

The generated 3D model data were imported into 3-matic (Materialise, Leuven, Belgium), followed by smoothing to generate a surface mesh. Subsequently, the mesh was imported into the ICEM CFD software (Ansys Inc, Canonsburg, PA, USA). The volume mesh of the airway had around 7 400 000 elements. The unstructured tetrahedral/prism hybrid mesh of the airway model was generated. Three layers of the prism mesh was placed near the wall so that even the area near the wall possessed sufficient resolution (Fig 3). The cell size of the prism region was adjusted to attain a dimensionless wall distance (y+) value less than 1.

### Airflow simulation

The above-mentioned analytical model was used to simulate function during inhalation. The conditions for analysis in this study were as follows: inhalation at rest at 20˚C and atmospheric

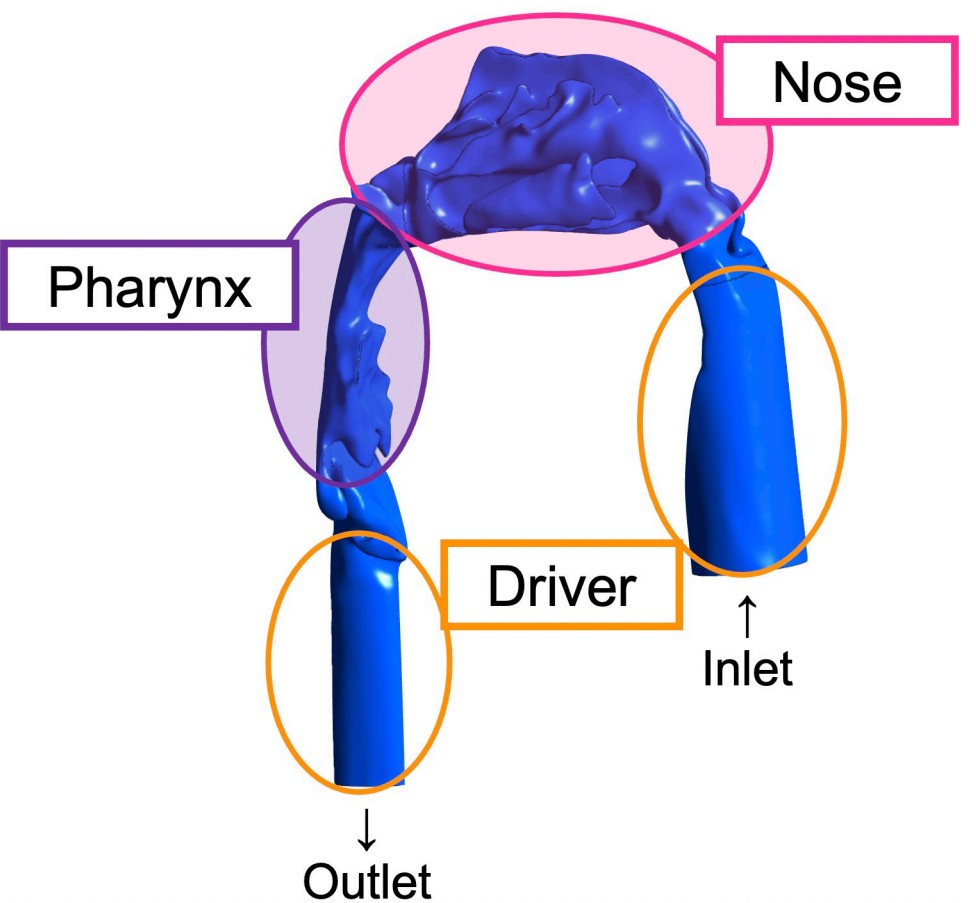

**Fig 2. Development of the driver.** Three-dimensional model of the nasal airway with the tubes projecting from the nostrils and subglottis indicating the "driver" region (encompassed by orange circles). The area surrounded by the pink oval is the nose area. The area surrounded by the purple oval is the pharynx area.

pressure ($1.013 \times 10^5$ Pa). The following physical properties were set in the model: steady flow of an incompressible Newtonian fluid with a density of 1.205 kg/m$^3$ and viscosity of $1.822 \times 10^{-5}$ Pa·s based on a previous study [26]. Lee et al. [26] explained that significant change was not observed in flow pattern distribution between steady and unsteady calculation at the inhalation phase.

The governing equations for the velocity and pressure of the flow field were solved using Fluent (version 14.0, ANSYS Inc., Canonsburg, PA, USA). The governing equations consist of the continuity Eq (1) and Navier-Stokes Eq (2) as follows.

Continuity equation:

$$\frac{\partial U_i}{\partial x_i} = 0 \tag{1}$$

Navier-Stokes equation:

$$\frac{\partial U_i}{\partial t} = -U_j \frac{\partial U_i}{\partial x_j} + \upsilon \frac{\partial^2 U_i}{\partial x_j^2} - \frac{1}{\rho} \left( \frac{\partial P}{\partial x_i} \right) + \frac{1}{\rho} \left( S_{F,i} \right) \tag{2}$$

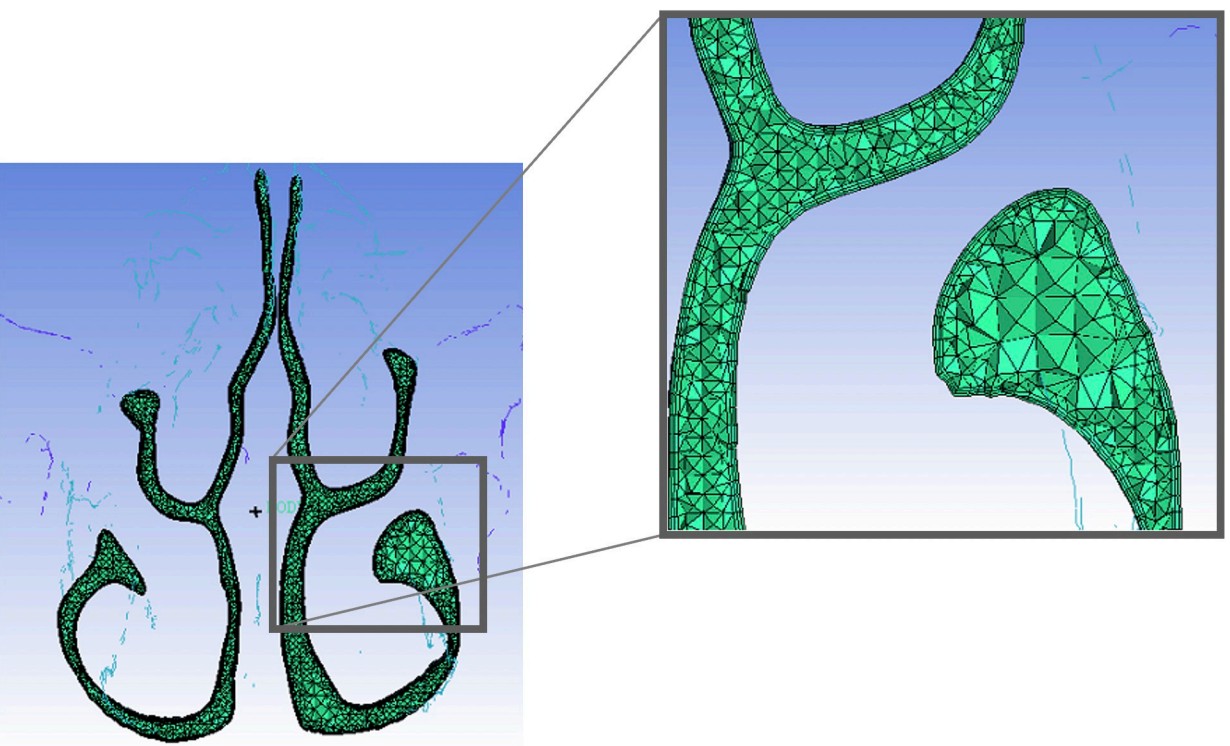

**Fig 3. Three-dimensional mesh cross-section of the nasal cavity.** The front section of the nasal cavity with a focus on the wall. The unstructured tetrahedral/prism hybrid mesh of the airway model was generated. Three layers of the prism mesh were placed near the wall.

Here, $U$ is the velocity vector, $i, j$ = 1, 2, 3, $U_i$ = $i$th component of the velocity vector, P = static pressure of the flow field, and $S_{F,i}$ = $i$th component of the source.

The finite volume method was used for the discretization of the governing equations. A semi-implicit method was used for time integration. The velocity and pressure fields were calculated using the SIMPLE algorithm (Semi-Implicit Pressure Linked Equation). The Launder-Sharma low Reynolds number k-ε model [27] was used as the turbulent flow model given by the following equations.

Turbulent kinetic energy equation (k):

$$\frac{\partial}{\partial t}(\rho k) + \frac{\partial}{\partial x_j}\left[\rho k u_j - \left(\mu + \frac{\mu_t}{\sigma_k}\right)\frac{\partial k}{\partial x_j}\right] = P - \rho\varepsilon - \rho D \tag{3}$$

Turbulence dissipation rate model (ε):

$$\frac{\partial}{\partial t}(\rho\varepsilon) + \frac{\partial}{\partial x_j}\left[\rho\varepsilon u_j - \left(\mu + \frac{\mu_t}{\sigma_\varepsilon}\right)\frac{\partial\varepsilon}{\partial x_j}\right] = (C_{\varepsilon 1}f_1 P - C_{\varepsilon 2}f_2\rho\varepsilon)\frac{\varepsilon}{k} + \rho E \tag{4}$$

$$\mu_t = \frac{C_\mu f_\mu \rho k^2}{\varepsilon} \tag{5}$$

$$P = \tau_{ij}^{turb}\frac{\partial u_i}{\partial x_j} \tag{6}$$

where $C_{\varepsilon 1}$, $C_{\varepsilon 2}$, $C_\mu$, $\sigma_k$ and $\sigma_\varepsilon$ are model constants. The damping functions $f_\mu$, $f_1$, and $f_2$ and the extra source terms D and E are only active close to the solid walls, which makes it possible to solve k and $\varepsilon$ down to the viscous sublayer. $f_\mu = \exp{-3.4/(1+Re_t/50)^2}$, $f_1 = 1$, $f_2 = 1-0.3\exp{-Re_t^2}$, $Re_t \equiv k^2/v\varepsilon$, $\varepsilon_{wall} = 0$, $D = 2v(\partial k^{1/2}/\partial y)^2$, $E = 2vv_t(\partial^2 u/\partial y^2)^2$. The constants appearing in (3), (4), (5), and (6) are $C_\mu = 0.09$, $\sigma_k = 1.0$, $\sigma_\varepsilon = 1.30$, $C_{\varepsilon 1} = 1.44$, and $C_{\varepsilon 2} = 1.92$, respectively.

The inlet boundary conditions were set at a flow rate of $2.000 \times 10^{-4}$ m$^3$/s based on previous studies on the peak respiratory flow at rest [28] and simulation of upper airway flow [29–31]. The inflow velocity was calculated using the flow rate and area of the inlet. The free outflow boundary condition was used as the outlet boundary condition. However, a pressure of P = 0 was used in the high-stenosis model with an inverse pressure gradient near the outlet. Because back flow occurred at the outflow boundary, the pressure boundary condition (P = 0) was adapted in the high-stenosis model based on reality. The wall was defined as non-slip.

## Areas and methods of evaluation

A simulation was performed using the model of the preoperative airway shape (PRE), which was developed using the preoperative CT data, and the model of the postoperative airway shape (POST), which was developed using CT data obtained 1 year after surgery. The effect of airway stenosis of the nasopharynx was examined using the 3D stenosis model (STENOSIS) with different amounts of trimming around the area extending from the POST to the PNS (Fig 4). In each model, the nasopharynx was trimmed as much as possible until it was divided into the nasal and laryngeal parts. As shown in Fig 4, the nasopharynx of the STENOSIS model (indicated by the rectangle in the inset) was trimmed mainly around the PNS by the amount indicated by the red asterisks (where each asterisk equals the amount of trimming for each model, e.g., in the STENOSIS -1 mm model, the asterisk means narrowing the thickness by 1 mm).

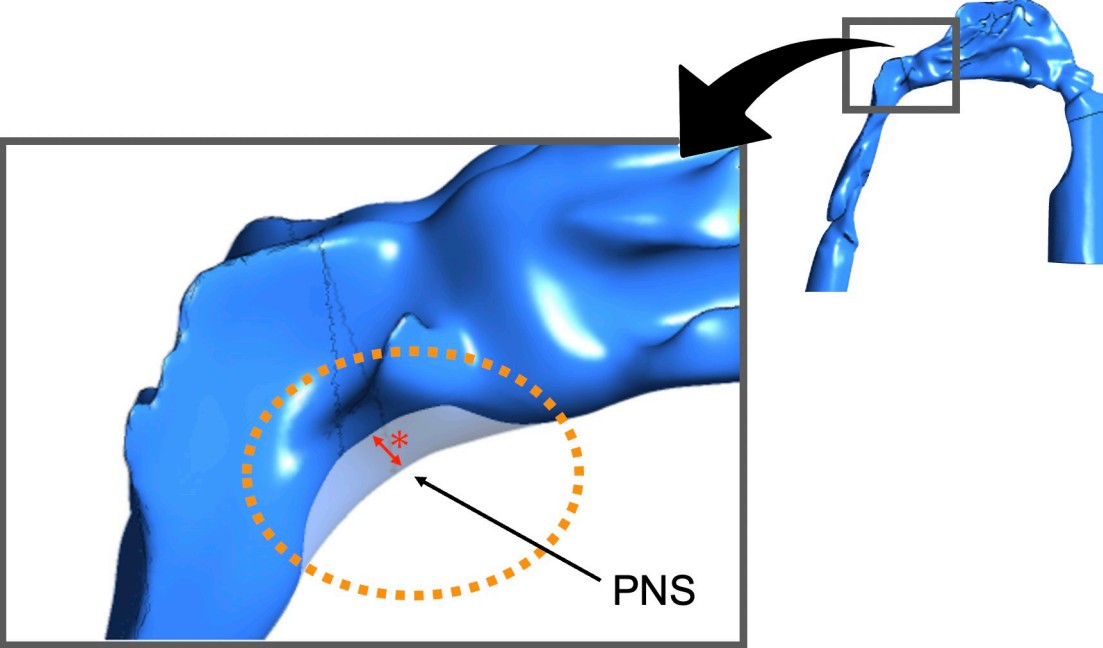

**Fig 4. STENOSIS model.** With a focus on the nasopharynx region of the STENOSIS model, the area (surrounded by the orange dotted circle) was trimmed by the length of the asterisk around the posterior nasal spine (PNS) in the sagittal plane. The red asterisk indicates the amount of trimming for the nasopharynx of the STENOSIS model.

Imaging simulation was performed using post-processing software (CFD-Post 14.0, ANSYS, Canonsburg, PA, USA). The generated morphologies and boundaries of the airway and sites of evaluation are shown in Fig 5. The inlet was perpendicular to the driver wall. The outlet was perpendicular to the subglottis wall. The nasal cavity was defined as the area extending from the nostril (Nos) to the posterior nasal aperture (PNA). The entity NP represents the cross-section of the flow crossing the PNS. $PA_t$ denotes the cross-section horizontal to the FH plane traversing the lower edge of the posterior nasal cavity. $PA_{min}$ denotes the narrowest part of the pharynx. $PA_t'$ is the cross-sectional area horizontal to the FH plane that divides the region extending from the $PA_t$ to $PA_{min}$ into two halves. $PA_b$ was defined as the horizontal plane crossing the tip of the epiglottis. The area extending from the PNA to the $PA_t'$ (mainly around the PNS) was trimmed in the STENOSIS model (Fig 4). The definitions of landmarks and measurement variables are listed in Tables 1 and 2.

Airway resistance was evaluated by measuring the pressure drop (ΔP), which was calculated by multiplying the airway resistance by the volumetric flow rate. A stable flow rate of $2.000 \times 10^{-4}$ m$^3$/s was maintained constant in this study. Therefore, airway resistance was proportional to the ΔP. ΔP was calculated as the difference in the mean pressure between two cross-sections obtained from the airway. Pressure drops in the nasal cavity and nasopharynx

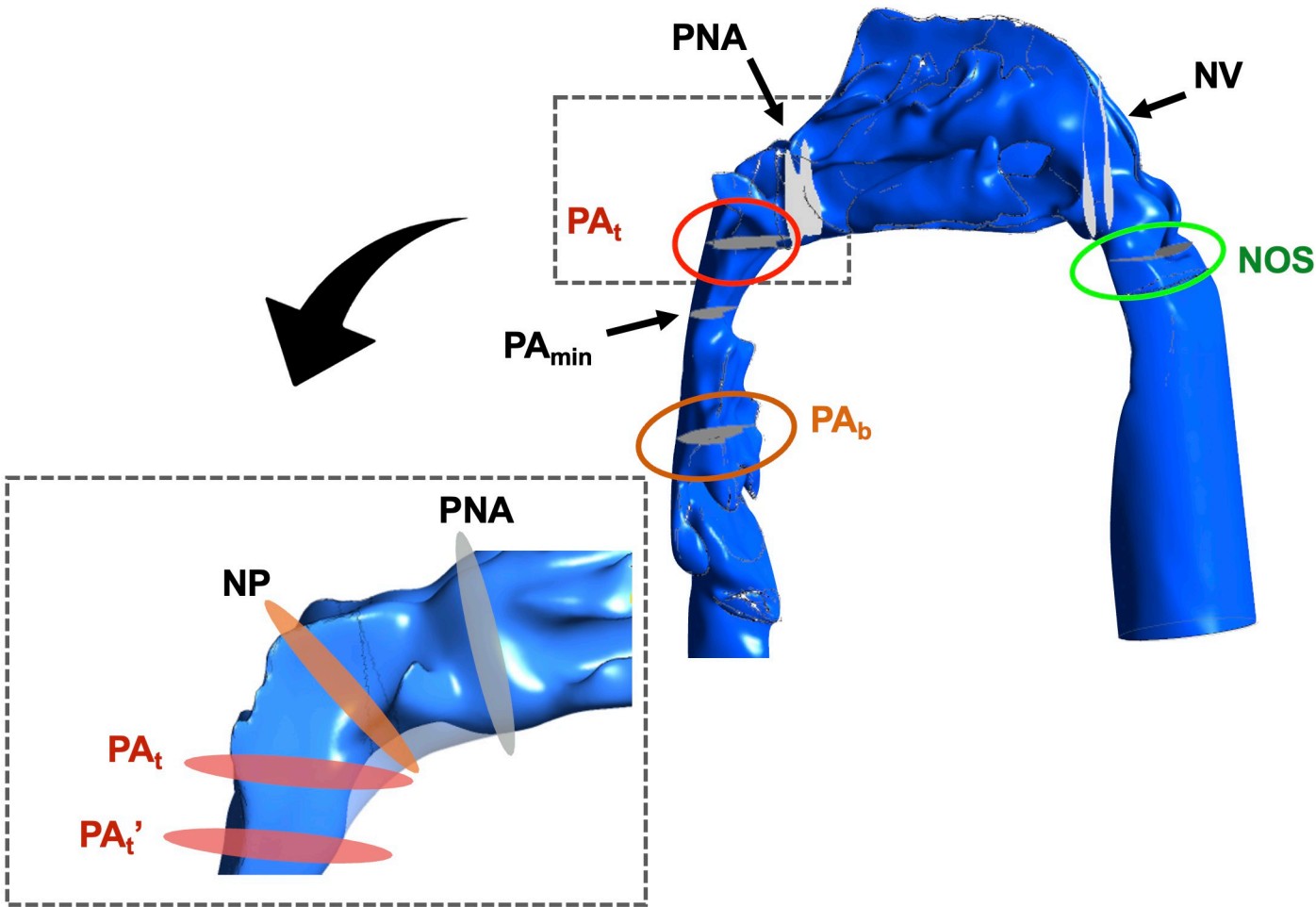

**Fig 5. Cross-section of the upper airway and nasopharynx.** Lateral view of the upper airway, nasopharynx and cross-section of the reference planes.

**Table 1. Definitions of landmarks.**

| Symbol | Definition |
|---|---|
| PNS | The cutting edge of the posterior nasal spine |
| U1 | The most anterior point on the incisal edge of the maxillary central incisor |
| U6 | The center of the occlusal surface of the maxillary first molar |
| FH | Frankfort horizontal plane |
| NOS | The nostrils |
| NV | The nasal valve |
| PNA | The posterior nasal aperture |
| CSA | The cross-sectional area of the upper airway on each CT sagittal plane |
| $\Delta P$ | The pressure drop |
| $\alpha$ | The rates of change in cross-sectional area |

were defined as $\Delta P_{Nose}$ (i.e., pressure drop from the NOS to the PNA) and $\Delta P_{Pharynx}$ (i.e., pressure drop from the PNA to the $PA_b$), respectively. $\Delta P_{All}$ (i.e., pressure drop from the NOS to the $PA_b$) was calculated as the sum of $\Delta P_{Nose}$ and $\Delta P_{Pharynx}$. The cross-sectional area (CSA) of the NOS, nasal valve (NV), PNA, NP, $PA_t$', $PA_{min}$, and $PA_b$ (as shown in Fig 5) was CSA-NOS, CSA-NV, CSA-PNA, CSA-NP, CSA-$PA_t$', CSA-$PA_{min}$, and CSA-$PA_b$, respectively. As shown in Fig 6, $\Delta P_{Pharynx}$ was divided into the following four segments:$\Delta P1$ (pressure drop from the PNA to the NP), $\Delta P2$ (pressure drop from the NP to the $PA_t$'), $\Delta P3$ (pressure drop from the $PA_t$' to the $PA_{min}$), and $\Delta P4$ (pressure drop from the $PA_{min}$ to the $PA_b$). The rates of change in the CSA of the mass extending from the nasopharynx to oropharynx ($\alpha$) were calculated as follows: $\alpha1$, CSA-NP /CSA-PNA; $\alpha2$, CSA-$PA_t$'/CSA-NP; $\alpha3$, CSA-$PA_t$'/CSA-$PA_{min}$; and $\alpha4$, CSA-$PA_{min}$/CSA-$PA_b$.

**Table 2. Definitions of measurements variables.**

| Symbol | Definition |
|---|---|
| PRE | three-dimensional pre-surgery model |
| POST | three-dimensional post-surgery model |
| STENOSIS | three-dimensional stenosis model |
| NP | cross-section of the flow crossing the PNS |
| $PA_t$ | cross-section horizontal to the FH plane crossing the lower edge of the posterior nasal cavity |
| $PA_{min}$ | the most constricted region of the pharyngeal airway |
| $PA_t$' | cross-section horizontal to the FH plane that divides the area extending from the $PA_t$ to $PA_{min}$ into two halves |
| $PA_b$ | horizontal plane crossing the tip of the epiglottis |
| CSA-x | cross-sectional area of each region of airway |
| $\Delta P_{All}$ | pressure drop in the whole upper airway |
| $\Delta P_{Nose}$ | pressure drop in the nasal cavity |
| $\Delta P_{Pharynx}$ | pressure drop in the nasopharynx |
| $\Delta P1$ | pressure drop from the PNA to the NP |
| $\Delta P2$ | pressure drop from the NP to the $PA_t$' |
| $\Delta P3$ | pressure drop from the $PA_t$' to the $PA_{min}$ |
| $\Delta P4$ | pressure drop from the $PA_{min}$ to the $PA_b$ |
| $\alpha1$ | calculated as CSA-NP /CSA-PNA |
| $\alpha2$ | calculated as CSA-$PA_t$'/CSA-NP |
| $\alpha3$ | calculated as CSA-$PA_t$'/CSA-$PA_{min}$ |
| $\alpha4$ | calculated as CSA-$PA_{min}$/CSA-$PA_b$ |

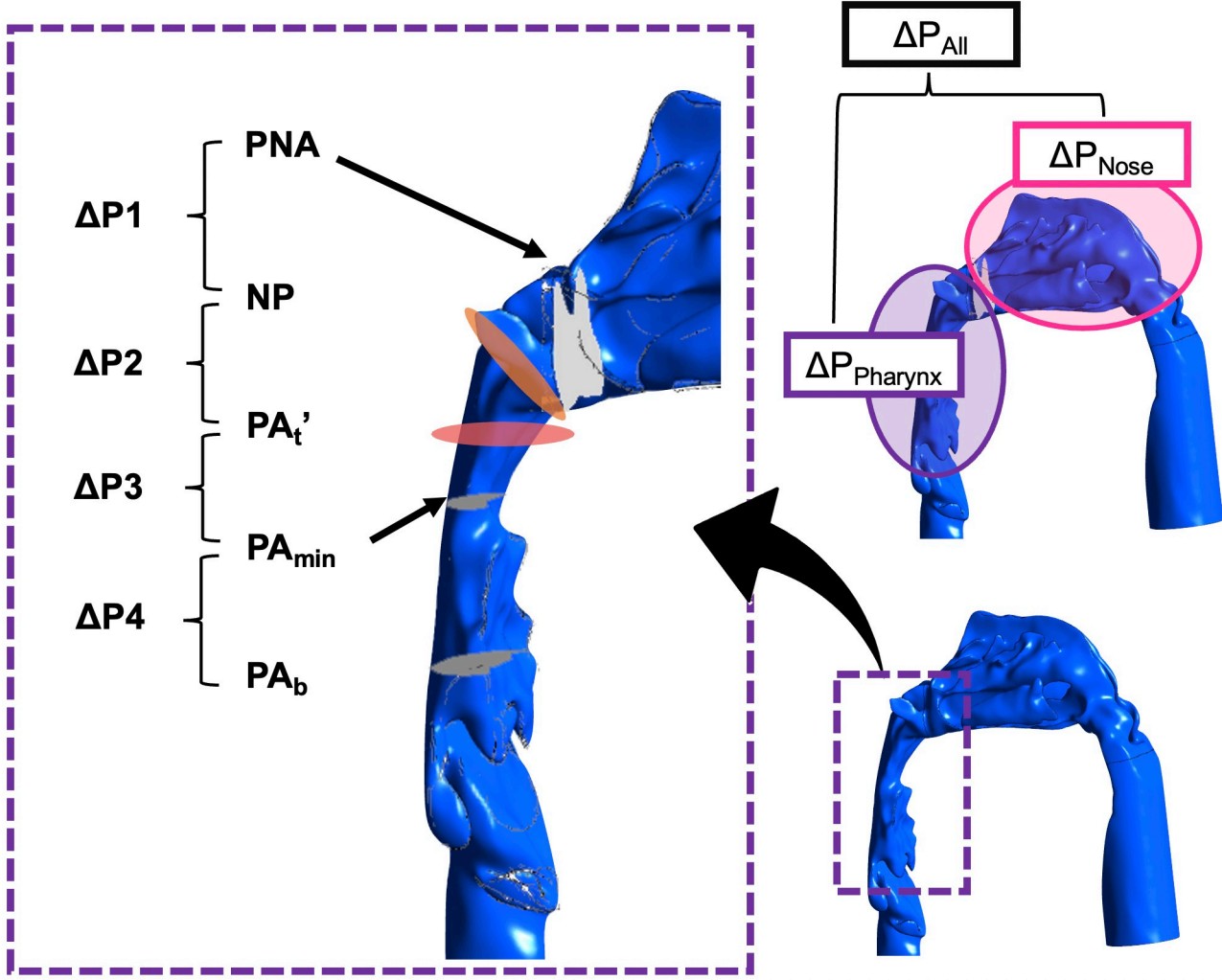

**Fig 6. Segments in which the pressure drop was evaluated.** Lateral view of the nasopharynx and oropharynx. Pressure drop was defined by the pressure at any two cross-sectional areas.

## Results

### Pressure effort

In the narrowest stenosis model (patients 1 and 2 in the STENOSIS model), $\Delta P_{All}$ was higher in POST than that in PRE (Table 3 and Fig 7). The ratio of $\Delta P_{Nose}$ to $\Delta P_{All}$ ($\Delta P_{Nose}/\Delta P_{All}$) was higher than that of $\Delta P_{Pharynx}$ to $\Delta P_{Nose}$ in all cases, except for patient 2 in the STENOSIS -10 mm model and patient 3 in the PRE model (Table 3 and Fig 7).

### Cross-sectional area

The comparison between PRE and POST showed that the CSA-NOS and CSA-Val were lower and CSA-PA$_{min}$ and CSA-PA$_b$ were higher in POST than those in PRE in all cases (Table 3). CSA PNA' and CSA-NP increased and decreased at different time points and under different conditions (Table 4). The results of the analyses of the shapes are depicted in Fig 8.

The rates of change in cross-sectional area ($\alpha$) were calculated as follows: $\alpha 1$, CSA-NP/CSA-PNA; $\alpha 2$, CSA-PA$_t$'/CSA-NP; $\alpha 3$, CSA-PA$_t$'/CSA-PA$_{min}$; and $\alpha 4$, CSA-PA$_{min}$/CSA-PA$_b$.

**Table 3. Pressure drop (ΔP).**

| Patient No. | Model | U1 posterior/vertical impaction (mm) | U6 | $\Delta P_{All}$ | $\Delta P_{Nose}$ (Pa) | $\Delta P_{Pharynx}$ (Pa) | $\Delta P_{Nose}/\Delta P_{All}$ | $\Delta P1+\Delta P2$ (Pa) | $\Delta P1$ (Pa) | $\Delta P2$ (Pa) | $\Delta P3+\Delta P4$ (Pa) | $\Delta P3$ (Pa) | $\Delta P4$ (Pa) |
|---|---|---|---|---|---|---|---|---|---|---|---|---|---|
| 1 | PRE | BMI:17.8 | | 11.318 | 9.009 | 2.309 | 0.80 | 0.485 | 0.033 | 0.452 | 1.823 | 2.800 | -0.977 |
| | POST | 2.5 / 4.0 | 3.0 / 5.5 | 9.179 | 8.953 | 0.226 | 0.98 | 0.023 | 0.068 | -0.045 | 0.203 | 0.550 | -0.347 |
| | STENOSIS | -1 mm | | 9.148 | 8.917 | 0.231 | 0.97 | 0.047 | 0.081 | -0.034 | 0.183 | 0.554 | -0.371 |
| | | -2 mm | | 9.197 | 8.922 | 0.275 | 0.97 | 0.139 | 0.137 | 0.002 | 0.136 | 0.478 | -0.342 |
| | | -3 mm | | 9.265 | 8.952 | 0.313 | 0.97 | 0.156 | 0.187 | -0.031 | 0.157 | 0.497 | -0.340 |
| | | -10 mm | | 10.901 | 9.445 | 1.456 | 0.87 | 1.772 | 1.780 | -0.008 | -0.317 | 1.018 | -1.335 |
| | | -15 mm | | 21.751 | 11.842 | 9.908 | 0.54 | 10.315 | 10.238 | 0.077 | -0.406 | 1.487 | -1.893 |
| 2 | PRE | BMI:19.4 | | 12.254 | 9.869 | 2.385 | 0.81 | 0.479 | 0.116 | 0.363 | 1.905 | 4.973 | -3.068 |
| | POST | 4.0 / 0 | 2.5 / 0 | 8.368 | 7.557 | 0.811 | 0.90 | 0.851 | -4.467 | 5.318 | -0.039 | 0.614 | -0.653 |
| | STENOSIS | -1 mm | | 8.509 | 7.594 | 0.915 | 0.89 | 0.866 | -3.716 | 4.582 | 0.049 | 0.640 | -0.591 |
| | | -2 mm | | 8.299 | 7.520 | 0.779 | 0.91 | 0.842 | -3.852 | 4.694 | -0.063 | 0.641 | -0.704 |
| | | -3 mm | | 8.417 | 7.600 | 0.817 | 0.90 | 1.055 | -4.259 | 5.314 | -0.238 | 0.567 | -0.805 |
| | | -5 mm | | 9.546 | 7.434 | 2.112 | 0.78 | 1.974 | -4.956 | 6.930 | -0.505 | 0.137 | -0.642 |
| | | -10 mm | | 28.291 | 8.004 | 20.287 | 0.28 | 20.486 | -6.523 | 27.009 | -0.517 | -0.199 | -0.318 |
| 3 | PRE | BMI:18.3 | | 31.591 | 15.387 | 16.204 | 0.49 | -0.036 | -0.178 | 0.142 | 16.241 | 9.624 | 6.617 |
| | POST | 7.5/2.2 | 8.0/2.5 | 28.729 | 22.728 | 6.001 | 0.79 | 0.034 | -0.163 | 0.197 | 5.968 | 5.315 | 0.653 |
| | STENOSIS | -1 mm | | 28.759 | 22.685 | 6.074 | 0.79 | 0.019 | -0.174 | 0.193 | 6.055 | 5.396 | 0.659 |
| | | -2 mm | | 28.647 | 22.695 | 5.952 | 0.79 | 0.020 | -0.084 | 0.104 | 5.933 | 5.296 | 0.637 |
| | | -3 mm | | 28.542 | 22.557 | 5.985 | 0.79 | 0.118 | -0.182 | 0.300 | 5.867 | 5.254 | 0.613 |
| | | -5 mm | | 28.289 | 22.864 | 5.425 | 0.81 | 0.142 | -0.200 | 0.342 | 5.984 | 5.282 | 0.702 |
| | | -10 mm | | 30.595 | 22.829 | 7.766 | 0.75 | 4.037 | 0.277 | 3.760 | 4.706 | 3.725 | 0.981 |

The relationship between ΔP (ΔP1+ΔP2 and ΔP3+ΔP4) and α for the four intervals of the nasopharynx in all models are shown in Fig 9 and the schematic is presented in Fig 10. The closer the α-value is to 1, the smaller the pressure drop irrespective of the area (Fig 9).

## Flow field

The flow field of the sagittal section of the nasopharynx for each model is shown in Fig 11. In the severe STENOSIS model, a jet was observed through the aperture stenosis, whereas vortices were observed downstream. In the STENOSIS -15mm model of patient 1 and the STENOSIS -10mm model of patient 2 in Fig 11, the Reynolds numbers of the nasopharynx region were about 11 200 and 6800, respectively.

## Discussion

A comparison of the PRE and POST models in this study revealed a lower $\Delta P_{All}$ in the POST model than in the PRE model (Fig 7 and Table 3). The airway resistance was proportional to ΔP because the flow rate was constant in this study. Therefore, the magnitude of ΔP was evaluated as the level of airway resistance, so the pre- and postoperative comparisons in this study revealed a postoperative improvement in the ventilation of the entire upper airway (Fig 7). In patient 3, $\Delta P_{Nose}$ was increased, but $\Delta P_{All}$, which represents the ΔP in the entire upper airway, was lower in the POST model than that in the PRE model. Our previous study [29] found that

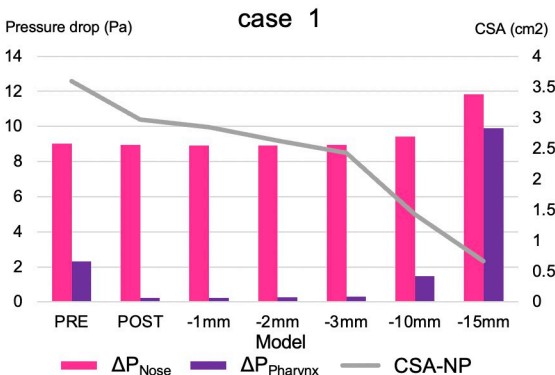

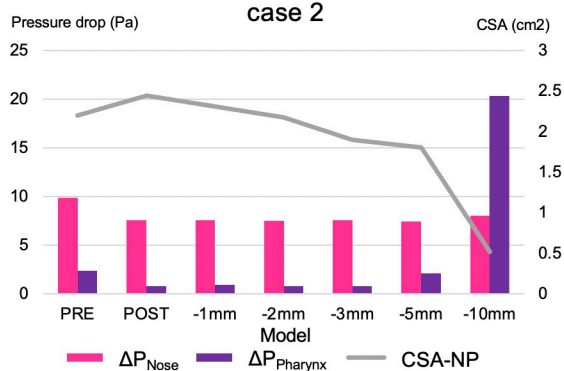

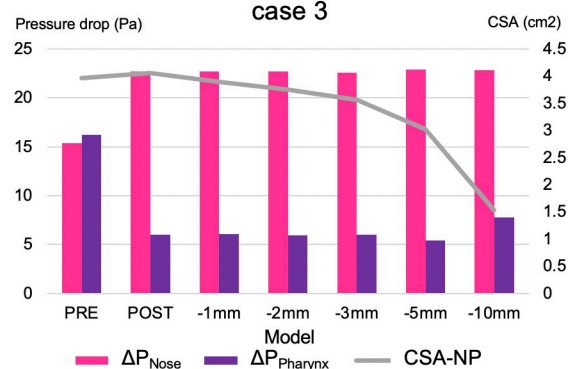

**Fig 7. Pressure drop in the nasal cavity and nasopharynx.** The line chart shows the CSA-NP. Abbreviations: -1 mm, STENOSIS -1 mm; -2 mm, STENOSIS -2 mm; -5 mm, STENOSIS -5 mm; -10 mm, STENOSIS -10 mm; -15 mm, STENOSIS -15 mm. Notes: pink bar, $\Delta P_{Nose}$; purple bar, $\Delta P_{Pharynx}$; gray line, CSA-NP.

the nasal cavity has a greater influence on the $\Delta P$ compared to the pharynx after surgery for mandibular prognathism (i.e., Class III skeletal relationship of the jaws). These findings are consistent with those of the present study, in which the ratio of $\Delta P_{Nose}$ to $\Delta P_{All}$ was higher (Table 3) than that of $\Delta P_{Pharynx}$ to $\Delta P_{All}$ in maxillary prognathism (i.e., Class II skeletal relationship of the jaws).

The results of our study suggest that changes in the nasopharynx do not have a substantial impact on the $\Delta P_{All}$ pressure drop, except in the extremely high STENOSIS model. The nasopharynx and oropharynx can be considered as a cylinder (Fig 8); thus, airway resistance is inversely proportional to the fourth power of the airway radius based on the Hagen–Poiseuille law (7) given by

$$Q = \frac{\pi \Delta P r^4}{8 \mu L} \tag{7}$$

where Q is the flow rate ($m^3$/s), $\Delta P$ is the pressure difference between the ends of the cylinder (Pa), r is the internal radius of the cylinder (m), μ is the viscosity of the fluid (Pa s), and L is the length of the cylinder (m). As the CSA of a cylinder is calculated by A = $\pi r^2$, $\Delta P$ is inversely proportional to the square of the CSA (i.e., $\Delta P = 8\pi\mu QL/A^2$).

The CSA of the nasopharynx (PNA, NP) is larger than that of the nasal cavity (NOS, NV) and the cross-sectional morphology of the nasopharynx closely resembles a cylindrical tube, while the nasal cavity is a narrow and complex structure. Thus, the nasopharynx has a lower

**Table 4. Cross-sectional area and ratio of the cross-sectional area.**

| Patient No. | Model | | CSA-NOS (cm$^2$) | CSA-NV (cm$^2$) | CSA-PNA′ (cm$^2$) | CSA-NP (cm$^2$) | CSA-PA$_t$' (cm$^2$) | CSA-PA$_{min}$ (cm$^2$) | CSA-PA$_b$ (cm$^2$) | α1 | α2 | α3 | α4 |
|---|---|---|---|---|---|---|---|---|---|---|---|---|---|
| 1 | PRE | 36 years 4 months | 1.323 | 2.016 | 3.388 | 3.592 | 2.125 | 0.963 | 2.038 | 1.06 | 0.59 | 0.45 | 2.12 |
| | POST | | 1.177 | 1.861 | 2.946 | 2.972 | 2.915 | 1.831 | 3.665 | 1.01 | 0.98 | 0.63 | 2.00 |
| | STENOSIS | -1 mm | 1.177 | 1.861 | 2.906 | 2.846 | 2.912 | 1.831 | 3.665 | 0.96 | 1.02 | 0.63 | 2.00 |
| | | -2 mm | 1.177 | 1.861 | 2.946 | 2.630 | 2.891 | 1.831 | 3.665 | 0.89 | 1.10 | 0.63 | 2.00 |
| | | -3 mm | 1.177 | 1.861 | 2.960 | 2.427 | 2.911 | 1.831 | 3.665 | 0.82 | 1.20 | 0.63 | 2.00 |
| | | -10 mm | 1.177 | 1.861 | 2.812 | 1.431 | 2.812 | 1.831 | 3.665 | 0.49 | 1.96 | 0.65 | 2.00 |
| | | -15 mm | 1.177 | 1.861 | 2.812 | 0.668 | 2.812 | 1.831 | 3.665 | 0.23 | 4.21 | 0.65 | 2.12 |
| 2 | PRE | 21 years 1 month | 1.069 | 1.936 | 2.985 | 2.201 | 2.222 | 0.674 | 2.047 | 0.74 | 1.01 | 0.30 | 0.92 |
| | POST | | 1.021 | 1.930 | 3.655 | 2.441 | 1.424 | 1.116 | 2.435 | 0.67 | 0.58 | 0.78 | 2.18 |
| | STENOSIS | -1 mm | 1.021 | 1.930 | 3.651 | 2.308 | 1.424 | 1.116 | 2.435 | 0.63 | 0.62 | 0.78 | 2.18 |
| | | -2 mm | 1.021 | 1.930 | 3.655 | 2.174 | 1.424 | 1.116 | 2.435 | 0.60 | 0.66 | 0.78 | 2.18 |
| | | -3 mm | 1.021 | 1.930 | 3.655 | 1.896 | 1.425 | 1.116 | 2.434 | 0.52 | 0.75 | 0.78 | 2.18 |
| | | -5 mm | 1.021 | 1.930 | 3.655 | 1.800 | 1.424 | 1.116 | 2.435 | 0.49 | 0.79 | 0.78 | 2.18 |
| | | -10 mm | 1.021 | 1.930 | 3.655 | 0.516 | 1.425 | 1.116 | 2.434 | 0.14 | 2.76 | 0.78 | 2.18 |
| 3 | PRE | 21 years | 1.238 | 1.901 | 4.001 | 3.964 | 3.768 | 0.584 | 1.328 | 0.99 | 0.95 | 0.16 | 2.27 |
| | POST | | 1.233 | 1.673 | 3.777 | 4.055 | 2.058 | 0.736 | 1.024 | 1.07 | 0.51 | 0.36 | 1.39 |
| | STENOSIS | -1 mm | 1.233 | 1.673 | 3.761 | 3.890 | 2.057 | 0.735 | 1.024 | 1.03 | 0.53 | 0.36 | 1.39 |
| | | -2 mm | 1.232 | 1.673 | 3.672 | 3.747 | 2.020 | 0.736 | 1.025 | 1.02 | 0.54 | 0.36 | 1.39 |
| | | -3 mm | 1.230 | 1.671 | 3.669 | 3.556 | 1.909 | 0.736 | 1.023 | 0.97 | 0.54 | 0.39 | 1.39 |
| | | -5 mm | 1.233 | 1.673 | 3.376 | 3.017 | 2.019 | 0.736 | 1.024 | 0.89 | 0.67 | 0.36 | 1.39 |
| | | -10 mm | 1.233 | 1.673 | 3.666 | 1.535 | 2.022 | 0.736 | 1.024 | 0.42 | 1.32 | 0.36 | 1.39 |

impact on ΔP in case of equivalent amount of jaw repositioning (Fig 8). Hence, it can be inferred that the effect of the nasopharynx on postero-superior repositioning of the maxilla is smaller than that of the nasal cavity. In the extremely high STENOSIS model, e.g., the STENOSIS -15 mm model of patient 1, when A in Eq (7) is less than one-fourth of that in POST, the calculated value of ΔP is larger than the square of 4. However, stenosis of such extreme severity does not occur clinically, owing to the presence of the descending palatine artery and pterygoid process, which regulate the postero-superior repositioning of the maxilla in Le Fort I osteotomy.

Note that in this study, we have assumed a rigid wall and steady state. The mechanical properties of the pharynx wall are difficult to determine, because it is regulated by a complex interplay between whether enclosed in a bony structure, wall thickness, airspace cross-sectional areas, and tissue pressure [32, 33]. Therefore, the compliance effects have not been considered and have been simplified. This behavior is an important aspect that should be taken in consideration in future studies. According to Hahn et al. [34], the upper airway wall can be assumed to be a rigid body for the purpose of simplification, ignoring the effects of the vibrissae, humidity, etc. Therefore, the ΔP, which mainly occurs in the upper airway, comprises a ΔP due to viscosity and ΔP due to turbulent flow. Air flow in the airway during breathing creates a resisting force due to the airflow viscosity in the upper airway. The presence of stenosis in the upper airway may lead to flow separation, which may lead to the formation of flow-separation zones as in the throat airflow structures [35, 36]. As shown in Fig 12, the flow velocity in the A-B

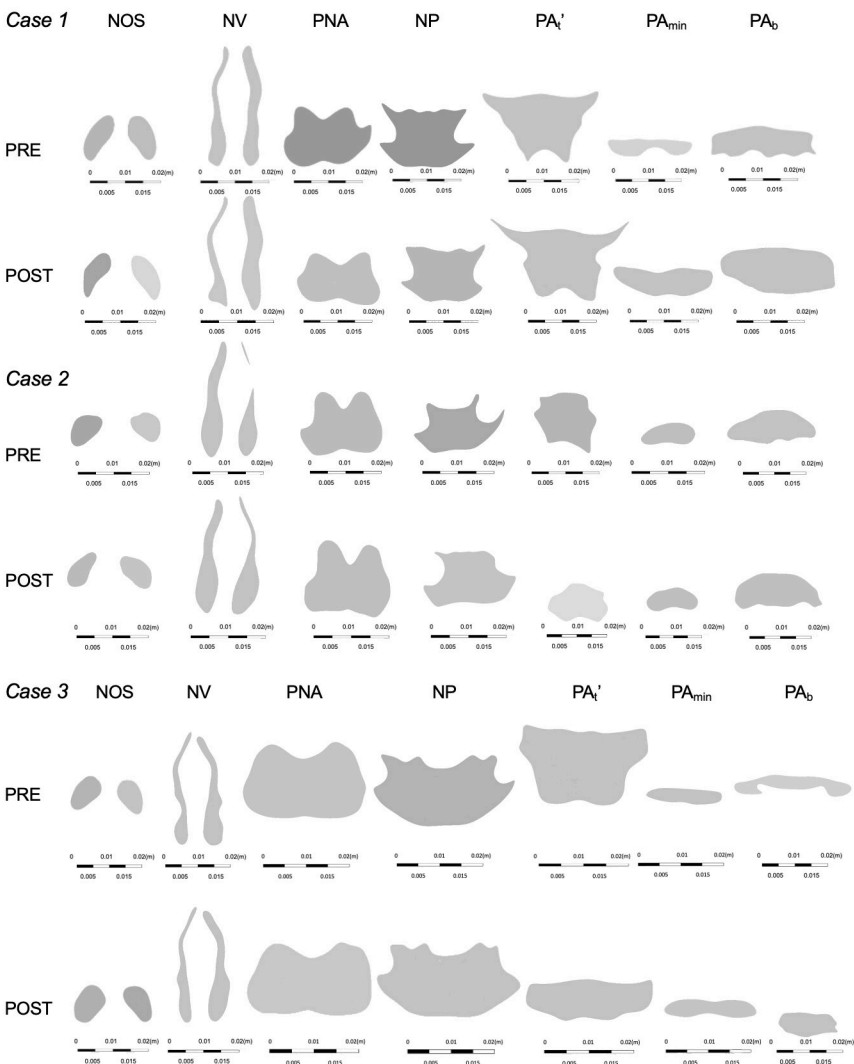

**Fig 8. Cross-sectional shape of the upper airway.** Cross-sectional shape of the upper airway in patients 1, 2 and patient 3.

interval is low due to a ΔP, whereas the flow velocity in the B-C interval is high due to an increase in the ΔP (albeit without flow-separation zones). This separation leads to a significantly higher amount of energy loss, in addition to airway resistance, depending on the size of the cross-section due to vortices created downstream of the separation point by reflux. According to the results of this study, separation and vena contracta in the narrower area (equivalent to the A-B interval in Fig 12) and separation and turbulent flow in a wider area (equivalent to the B-C interval in Fig 12) resulted in a ΔP due to turbulent flow in $\Delta P_{Pharynx}$, as seen in the STENOSIS -15 mm model of patient 1 and STENOSIS -10 mm model of patient 2 in Fig 11. On the other hand, the changes in ΔP were insignificant due to a significant ΔP caused by viscosity due to the tapering of the pharyngeal airway toward the narrowest part of the pharynx in the POST/STENOSIS models of patients 2 and 3. Previous studies [30, 37] examined the CSA of stenosis of the upper airway, but failed to evaluate the changes in the pre- and postoperative diameters of the upper airway. Yajima et al. [30] found that the ΔP increased

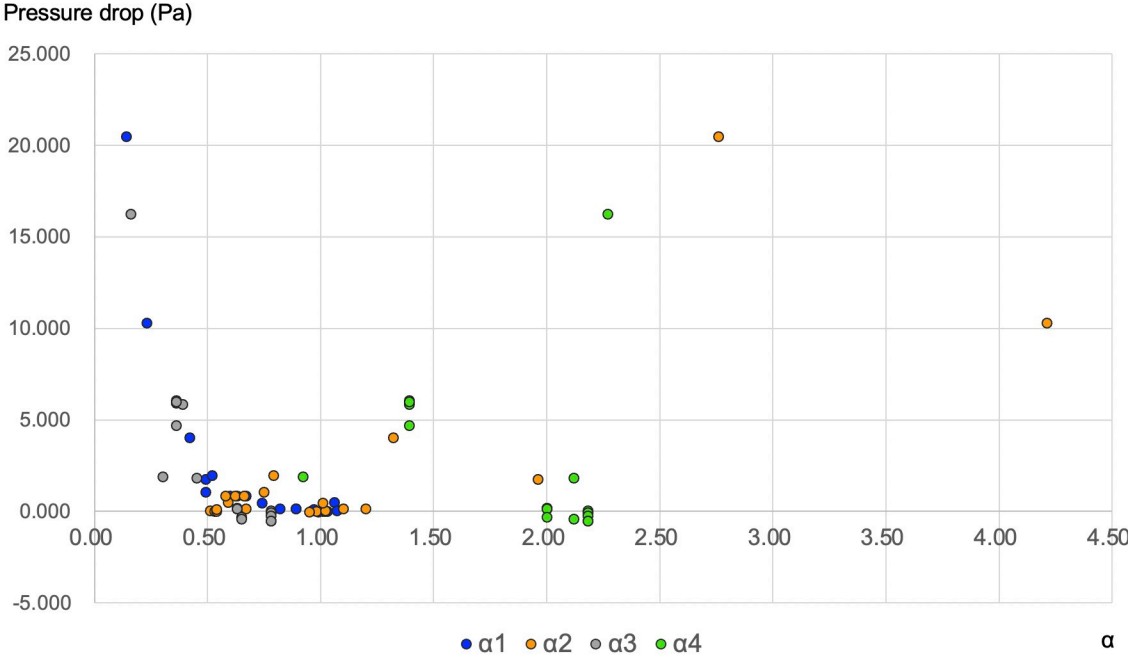

Pressure drop (Pa)

**the rate of changes in the cross-sectional area (α)**

α1: CSA-NP /CSA-PNA    α2:CSA-PA$_t$'/ CSA-NP    α3: CSA-PA$_t$'/CSA-PA$_{min}$    α4: CSA-PA$_{min}$/CSA-PA$_b$

**Fig 9. Pressure drop and the rate of changes in the cross-sectional area (α).** Correlation between pressure drop and α. The horizontal axis indicates α and the vertical axis represents the pressure drop. Notes: blue, α1; orange, α2; grey, α3; green, α4.

substantially when the stenotic region in the oropharynx (CSA-PA$_{min}$) was less than 1 cm$^2$. Conversely, for the nasopharynx (CSA-NP), ΔP increased below 1 cm$^2$ (Table 4 and Fig 7). The rate of changes in the CSA (α) was calculated to facilitate objective comparisons between the pre- and postoperative states using ΔP (Figs 9 and 10). The greater the proximity of the value of α to 1, the smaller the changes in airway diameter and ΔP. On the other hand, the

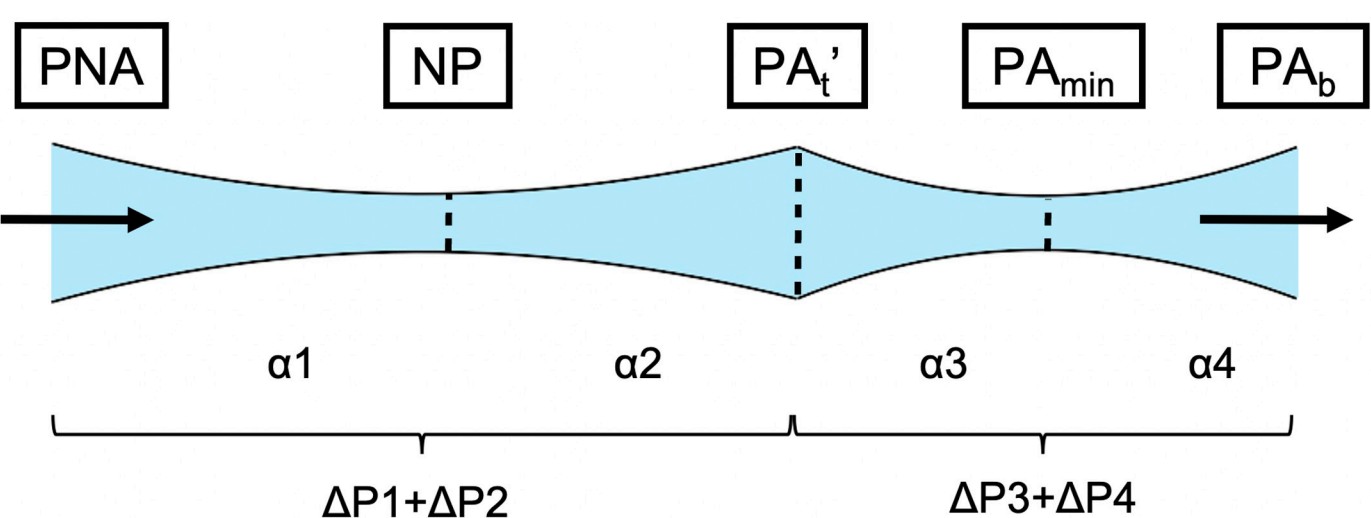

**Fig 10. Schematic illustration of the pharynx.** The arrow points toward the direction of airflow.

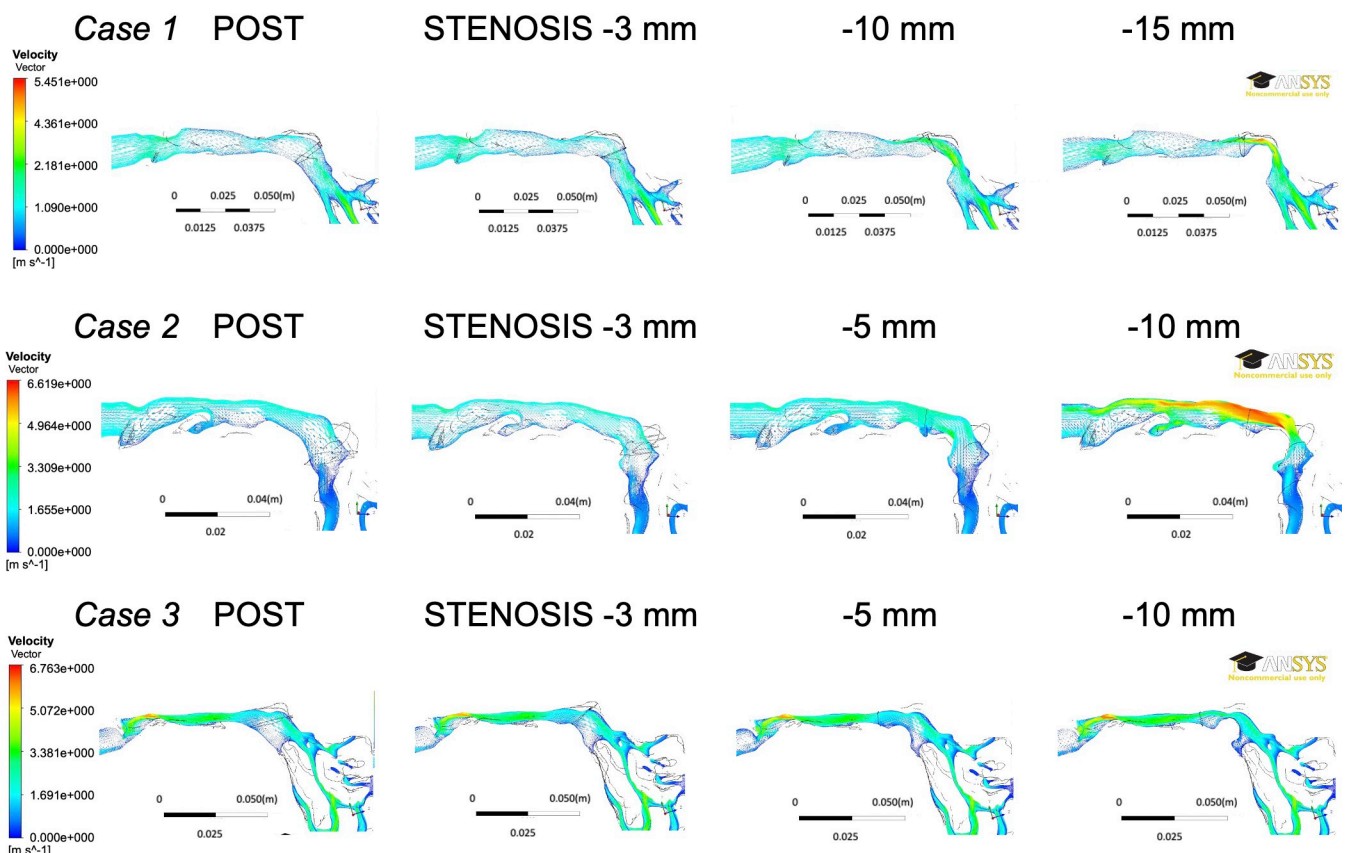

**Fig 11. Flow fields of the sagittal section of the nasopharynx for each model.** The direction of flow is from the lower right (nasal cavity side) to the left (oropharyngeal side).

further the value of α from 1, the greater the changes in airway diameter and pressure drop. Consider the α2 of the STENOSIS model of patient 1: -1 mm and -2 mm are both close to 1,

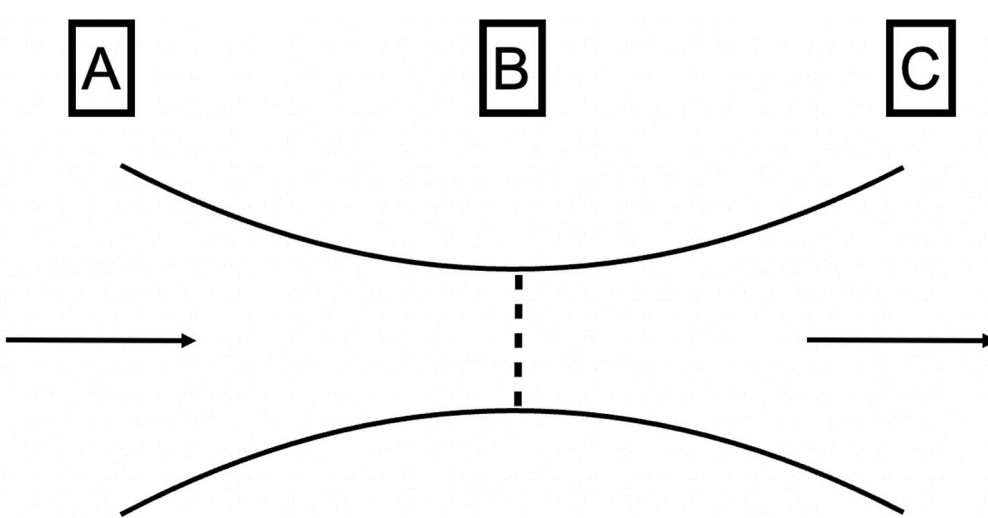

**Fig 12. Stenosis image.** The arrow points toward the direction of airflow. Notes: A, inlet; B, stenosis; C, outlet.

$\alpha2$ is $> 1$, and the pressure drop is also higher in the -10-mm and -15-mm models. These results suggest that the closer the morphology of the airway is to a straight tube, the lower the risk of reduction of airway ventilation. Therefore, ideally the airway diameter should be constant. In this study, the $\alpha$-value and CSAs were sufficient, so that all values of $\Delta P_{All}$ in the POST model were smaller than their PRE counterparts and there was no reduction in nasal respiratory function.

The prediction of the changes in airway morphology due to the repositioning of bone fragments may be necessary, to apply the present findings of the airway morphology to surgical practice in the future. It is thought that the direction of repositioning of the maxilla during corrective jaw surgery affects the nasopharyngeal airway [38], whereas repositioning of the mandible affects the lower part of the pharyngeal airway [39]. However, the sample size of this study was small because the indications for maxillary osteotomy alone are few, so larger sample sizes and further analyses are needed to accurately predict changes in the airway morphology, which varies according to the maxillofacial morphology and amount of surgical repositioning. We will endeavor to investigate this aspect in a future study. If it becomes clear that there are no functional problems with surgical methods that involve the large posterior and/or superior movement of the jawbone, which is expected to have an adverse effect on the upper airway, it will be possible to improve surgical planning and develop new treatments. This would lead to the enhancement of patients' quality of life and the further development of orthognathic surgery. Individual differences exist in the morphological changes in the airway even after equivalent amount of repositioning. Furthermore, a reduction in the airway CSA may lead to stenosis, while flattening of the airway may reduce the anteroposterior diameter, also leading to stenosis. However, the patterns of airway changes have not been elucidated. Therefore, in the present study, we added fluid considerations from a case study with a small sample size by mimicking the airway constriction caused by jaw movement and changing it numerically. What we learned from that consideration is highly versatile as it can be applied to other patients if the relationship holds. Elucidation of the airway morphology and prediction of the changes in respiratory function using preoperative CT in future studies, along with the findings of the present study, may aid surgical planning, with considerations for occlusion, maxillofacial morphology, and respiratory function. Not only can we develop surgical methods to prevent the onset of OSA, but we can develop clinical research that incorporates model simulations of soft tissue changes including the upper airway associated with general orthognathic surgery.

The current study of the effect of postero-superior repositioning of the maxilla on nasal ventilation using computational fluid dynamics showed that current surgical methods are appropriate with respect to functionality and do not reduce nasal respiratory function, irrespective of the nature of the surgery. The greater the proximity of the value of $\alpha$ to 1, the smaller the changes in airway diameter and $\Delta P$, so ideally the airway diameter should be constant without stenosis. This study identified airway shapes that are preferable from the perspectives of flow dynamics. We found that our results may be applicable to other patients.

## Author Contributions

**Conceptualization:** Misaki Aoyagi, Soma Kita, Takashi Ono.

**Data curation:** Misaki Aoyagi, Koichi Fujita, Haruki Imai.

**Formal analysis:** Misaki Aoyagi, Marie Oshima, Masamichi Oishi.

**Funding acquisition:** Soma Kita, Shuji Oishi.

**Investigation:** Misaki Aoyagi, Soma Kita.

**Methodology:** Misaki Aoyagi, Soma Kita.

**Project administration:** Misaki Aoyagi.

**Resources:** Koichi Fujita, Haruki Imai.

**Software:** Misaki Aoyagi, Soma Kita.

**Supervision:** Hiroko Ohmori, Takashi Ono.

**Validation:** Marie Oshima, Masamichi Oishi.

**Visualization:** Misaki Aoyagi.

**Writing – original draft:** Misaki Aoyagi.

**Writing – review & editing:** Misaki Aoyagi, Marie Oshima, Masamichi Oishi, Soma Kita, Koichi Fujita, Haruki Imai, Shuji Oishi, Hiroko Ohmori, Takashi Ono.

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
