## [Decision Letter · Decision Letter 0]

12 Jan 2022

PONE-D-21-36984Computational fluid dynamic analysis of the nasal respiratory function before and after postero-superior repositioning of the maxillaPLOS ONE

Dear Dr. Oshima,

Thank you for submitting your manuscript to PLOS ONE. After careful consideration, we feel that it has merit but does not fully meet PLOS ONE’s publication criteria as it currently stands. Therefore, we invite you to submit a revised version of the manuscript that addresses the points raised during the review process.

We look forward to receiving your revised manuscript.

Kind regards,

Shaokoon Cheng

Academic Editor

PLOS ONE

Journal Requirements:

Reviewers' comments:

**Comments to the Author**

1. Is the manuscript technically sound, and do the data support the conclusions?

Reviewer #1: Partly

Reviewer #2: Partly

2. Has the statistical analysis been performed appropriately and rigorously? 

Reviewer #1: N/A

Reviewer #2: N/A

3. Have the authors made all data underlying the findings in their manuscript fully available?

Reviewer #1: Yes

Reviewer #2: Yes

4. Is the manuscript presented in an intelligible fashion and written in standard English?

Reviewer #1: Yes

Reviewer #2: Yes

5. Review Comments to the Author

Reviewer #1: 1. The significance of the study is not clear, and it should be explained better in the Abstract, Objectives, as well as the Discussion/Conclusion.

2. The authors explain that the sample size for the study has been small and they tend to study a larger sample size in future. Adding more explanations on how the results of this study or the knowledge obtained in this research can facilitate future research or bring insights into an efficient future work can be helpful.

3. In the abstract, the observation of “ΔP in the upper airway was lower in the POST model compared to the PRE model” can be more elaborated before concluding that the current surgical methods “do not compromise nasal respiratory function”. For instance, why the change in pressure drop cannot make any change on respiratory function? Could it not affect the air flow rate? Or, what changes were they expecting that were not observed?

4. In the Abstract and Discussion, it is mentioned that “this study identified airway shapes that are preferable from the perspectives of flow dynamics”. However, the text does not provide explanations about this identification and it is not mentioned in the text what criteria are considered for evaluating an airway shape as favourable from fluid dynamics perspective. It will be good if authors provide more explanations on that.

5. The first paragraph of the introduction requires references (lines 43-45).

6. The purpose of the lines 68-69 is not clear and the grammatical structure requires amendment.

7. Authors may clarify figure 1 by improving the image quality and showing the rotations with arrows and probably the position of the nasal cavity and nasopharyngeal airway.

8. In line 134, the atmospheric pressure mentioned is inaccurate.

9. All the terms such as Pamin, Pabp, Paup, and CSA-PNA are recommended to be denoted in a more clear and readable way.

10. All figure captions should be re-written in the form of complete sentence(s) and not phrases, where possible. Also, figure captions are better to be written as one single paragraph and not multiple paragraphs.

11. All parameters studied in the model can be presented in a nomenclature in the beginning or a table in the text to avoid repeating the definitions in every figure caption and at the end of each Table. For instance, alpha is defined in both lines 261 and 270 repeatedly.

12. In Table 1, units should be mentioned in front of each parameter rather than in separate cells.

13. The quality of all figures should be improved.

14. Table 1 should be cross-referenced in line 243, where the data is reported.

15. In figure 8, appropriate labels should be added to both X and Y axes instead of explaining in the figure caption.

16. The statement in lines 264-265 needs to be edited. Also, Figure 9 should be cross-referenced as the data is reported in lines 264-265.

17. The two statements in lines 301 to 304 give the same message and hence, can be merged into one sentence.

18. The statement in lines 299-300 seems to disagree with the statement in lines 310-312. Please clarify.

19. The statements in lines 329-331 need references.

20. In line 337, please specify how it was found that the flow was turbulent. Please report the Re number if that is used.

Reviewer #2: The manuscript offers good insights into the effect of changes in airway morphology due to the repositioning of bone fragments which is beneficial in predicting the effect of such operation on patients ventilation, but the number of cases used in the study is quite small which makes it quite hard to get more generalized conclusions. However, the study approach is quite promising for the future of corrective surgeries.

- In the abstract, page (2) line (32-33),” the rate of change in the cross-sectional area of the mass extending from the nasopharynx to oropharynx approximated 1”. It is not clear how the rate of change in the cross-sectional area was calculated?

- In the second paragraph in the introduction, page (4) line (55-69), this paragraph is a bit confusing to the reader, as it is not clear what the authors are trying to address in this paragraph and how it is related to the current work.

- Figure (1) need to be more intuitive by adding a color legend to address reactionary counter-clockwise rotation during postero-superior repositioning.

- typo error line (90) page (6),” Surgical impaction of the maxilla and the reaction of the mandible ae illustrated schematically”.

- It has to be mentioned clearly that the study was performed on females only because the authors mentioned that the study was performed on 3 females.( page 7, line 100).

- Figure illustrating the mesh is needed to provide information regarding this information “Three layers of the tetrahedral/hybrid tetrahedral-prism”, mentioned in line 129, page(8).

- Information about the element size and mesh independence study need to be added.

- Reason for choosing these assumptions is steady and need to be clarified in the text. Line (135), page (9).

- In page (10) line (162-163), the Outlet boundary condition is used a free outlet and p=0 in one of the cases, can you clarify the reason behind using different outlet boundary conditions.

- Figure (6), a y-axis label need to be added and the x-axis.

- Can you indicate how the STENOSIS -1mm, -3mm is measured in the figure (3), it seems to be a distance, so can you clarify how this is measured

- In page (25), line (327), the assumption of rigid upper airways is quite obsolete; as some studies have investigated the effect of upper airways tissue motion (The effects of upper airway tissue motion on airflow dynamics). Also, most of the experimental studies use flexible materials for manufacturing upper airway replicas’.

- Body mass index (BMI) for the cases tested in this study needs to be mentioned.

6. PLOS authors have the option to publish the peer review history of their article (what does this mean?). If published, this will include your full peer review and any attached files.

Reviewer #1: No

Reviewer #2: No

---

## [Author Response · Author response to Decision Letter 0]

25 Feb 2022

Here is a point-by-point response to the journal requirements and the reviewers’ comments. 

Journal Requirements 

1. When submitting your revision, we need you to address these additional requirements.Please ensure that your manuscript meets PLOS ONE's style requirements, including those for file naming. The PLOS ONE style templates can be found at 

Reply: 

We have ensured that our manuscript meets PLOS ONE's style requirements, including those for file naming.

Reply: 

We have ensured that our reference list is complete and correct. Additional references have been noted in specific replies and as follows:

1. Tselnik M, Pogrel MA. Assessment of the pharyngeal airway space after mandibular setback surgery. J Oral Maxillofac Surg. 2000;58: 282–5 discussion 285. doi: 10.1016/s0278-2391(00)90053-3 (line 42).

2. Mirmohamadsadeghi H, Zanganeh R, Barati B, Tabrizi R. Does maxillary superior repositioning affect nasal airway function? Br J Oral Maxillofac Surg. 2020;58: 807-811 doi: 10.1016/j.bjoms.2020.04.020 (line 42).

3. Engboonmeskul T, Leepong N, Chalidapongse P. Effect of surgical mandibular setback on the occurrence of obstructive sleep apnea. J Oral Biol Craniofac Res. 2020;10: 597-602 doi: 10.1016/j.jobcr.2020.08.008. (line 43).

4. Hsieh YJ, Chen YC, Chen YA, Liao YF, Chen YR. Effect of bimaxillary rotational setback surgery on upper airway structure in skeletal class III deformities. Plast Reconstr Surg. 2015;135: 361e-369e doi: 10.1097/PRS.0000000000000913 (line 43).

32. White DP, MK Younes. Obstructive sleep apnea. Compr Physiol. 2012;2: 2541–2594 doi: 10.1002/cphy.c110064 (line 309).

33. Woodson BT. A method to describe the pharyngeal airway. Laryngoscope 2015;125: 1233–1238 doi: 10.1002/lary.24972. (line 309).

35. Kleinstreuer C, Zhang Z. Airflow and particle transport in the human respiratory system. Annu Rev Fluid Mech. 2010; 42: 301–334 doi: 10.1146/annurev-fluid-121108-145453 (line 316).

36. Cui XG, Gutheil E. Large eddy simulation of the unsteady flow-field in an idealized human mouth-throat configuration. J Biomech. 2011; 44: 2768–2774 doi: 10.1016/j.jbiomech.2011.08.019 (line 316).

3. We note that you have indicated that data from this study are available upon request. PLOS only allows data to be available upon request if there are legal or ethical restrictions on sharing data publicly. For more information on unacceptable data access restrictions, please see

http://journals.plos.org/plosone/s/data-availability#loc-unacceptable-data-access-restrictions. 

b) If there are no restrictions, please upload the minimal anonymized data set necessary to replicate your study findings as either Supporting Information files or to a stable, public repository and provide us with the relevant URLs, DOIs, or accession numbers. For a list of acceptable repositories, please see 

http://journals.plos.org/plosone/s/data-availability#loc-recommended-repositories.

Reply: 

a) There is an ethical restriction. Upon reasonable request, the data that support the findings of this study are available from the ethics committee as follows:

Dental Research Ethics Committee of Tokyo Medical and Dental University

E-mail: d-hyoka.adm@tmd.ac.jp

Url: https://tmdu-berc.jp

Comments from Reviewer #1 

We wish to express our appreciation to the reviewer for your insightful comments, which have helped us significantly improve the paper.

1. The significance of the study is not clear, and it should be explained better in the Abstract, Objectives, as well as the Discussion/Conclusion.

Reply: 

In accordance with the reviewer's comment, we have added the explanation in the manuscript as follows:

“In particular, nasopharyngeal stenosis, because of the complex influence of both jaws, the effects of which have not yet been clarified owing to postero-superior repositioning of the maxilla may significantly impact sleep and respiratory function, necessitating further functional evaluation.” (in the Abstract, lines 18-21)

“The jawbone and the soft tissues and airway that are moved during orthognathic surgery can be associated with all of these source sites because of their location.” (in the Introduction, lines 50-51)

“Therefore, in the present study, we added fluid considerations from a case study with a small sample size by mimicking the airway constriction caused by jaw movement and changing it numerically. What we learned from that consideration is highly versatile as it can be applied to other patients if the relationship holds.” (in the Discussion, lines 356-358)

“We found that our results may be applicable to other patients.” (in the Discussion section, conclusion, line 369)

2. The authors explain that the sample size for the study has been small and they tend to study a larger sample size in future. Adding more explanations on how the results of this study or the knowledge obtained in this research can facilitate future research or bring insights into an efficient future work can be helpful.

Reply: 

In accordance with the reviewer's comment, we have added the explanation in the manuscript as follows:

“The new insights acquired in this study improves understanding of the pathogenesis of OSA and the effect of orthognathic surgery.” (in the Introduction, lines 94-95)

“If it becomes clear that there are no functional problems with surgical methods that involve the large posterior and/or superior movement of the jawbone, which is expected to have an adverse effect on the upper airway, it will be possible to improve surgical planning and develop new treatments. This would lead to the enhancement of the patients’ quality of life and the further development of orthognathic surgery.” (in the Discussion, lines 349-352)

“Therefore, in the present study, we added fluid considerations from a case study with a small sample size by mimicking the airway constriction caused by jaw movement and changing it numerically. What we learned from that consideration is highly versatile as it can be applied to other patients if the relationship holds.” (in the Discussion, lines 356-358)

“Not only can we develop surgical methods to prevent the onset of OSA, but we can develop clinical research that incorporates model simulations of soft tissue changes including the upper airway associated with general orthognathic surgery.” (in the Discussion, lines 361-363)

3. In the abstract, the observation of “ΔP in the upper airway was lower in the POST model compared to the PRE model” can be more elaborated before concluding that the current surgical methods “do not compromise nasal respiratory function”. For instance, why the change in pressure drop cannot make any change on respiratory function? Could it not affect the air flow rate? Or, what changes were they expecting that were not observed?

Reply: 

As reported by Mirmohamadsadeghi H et al. [2] and Hsieh YJ et al. [4], nasal airflow, cross-sectional area, and volume are decreased by maxillary movement. Therefore, in the present study, it was feared that postoperative orthognathic surgery with postero-superior repositioning of the maxilla would result in a larger ΔPAll due to these decreases, which would worsen the respiratory function. However, according to the results, ΔPAll in the upper airway was lower in the POST model than in the PRE model. As shown in Fig. 8, this can be explained by focusing on the change in airway cross-section.

We have therefore changed and added the following text:

From

“This study aimed to perform a functional evaluation of the effects of corrective jaw surgery involving postero-superior repositioning of the maxilla on nasal respiratory function using computational fluid dynamics for treatment planning.”

to

“This study aimed to perform a functional evaluation of the effects of surgery involving maxillary repositioning, which may result in a larger airway resistance due to the stenosis would worsen the respiratory function, using CFD for treatment planning.” (in abstract, lines 21-23)

“A comparison of the PRE and POST models in this study revealed a lower ΔPAll in the POST model than in the PRE model (Fig. 7, Table 3).” (in discussion, lines 280-281)

4. In the Abstract and Discussion, it is mentioned that “this study identified airway shapes that are preferable from the perspectives of flow dynamics”. However, the text does not provide explanations about this identification and it is not mentioned in the text what criteria are considered for evaluating an airway shape as favourable from fluid dynamics perspective. It will be good if authors provide more explanations on that.

Reply: 

In accordance with the reviewer's comment, we have added the explanation in the Abstract as follows:

“The closer the α-value is to 1, the smaller the ΔP, so ideally the airway should be constant.” (line 34)

Also, we have added the explanation in the Discussion as follows:

“The greater the proximity of the value of α to 1, the smaller the changes in airway diameter and ΔP, so ideally the airway diameter should be constant without stenosis.” (in discussion, lines 366-368)

5. The first paragraph of the introduction requires references (lines 43-45).

Reply: 

We have added references [1,2] (line 42) and [3,4] (line 43) in the paragraph. 

The introduction section of the manuscript now contains the following references: 

1. Tselnik M, Pogrel MA. Assessment of the pharyngeal airway space after mandibular setback surgery. J Oral Maxillofac Surg. 2000;58: 282–5

2. Mirmohamadsadeghi H, Zanganeh R, Barati B, Tabrizi R. Does maxillary superior repositioning affect nasal airway function? Br J Oral Maxillofac Surg. 2020;58: 807-811.

3. Engboonmeskul T, Leepong N, Chalidapongse P. Effect of surgical mandibular setback on the occurrence of obstructive sleep apnea. J Oral Biol Craniofac Res. 2020;10: 597-602.

4. Hsieh YJ, Chen YC, Chen YA, Liao YF, Chen YR. Effect of bimaxillary rotational setback surgery on upper airway structure in skeletal class III deformities. Plast Reconstr Surg. 2015;135: 361e-369e. 

6. The purpose of the lines 68-69 is not clear and the grammatical structure requires amendment.

Reply: 

We are sorry that this part was not clear in the original manuscript. We have revised the contents of this part from:

“Prevention the reduction in overall ventilation of the upper airway necessitates the evaluation of the nasopharynx, the site of the original stenosis.”

to

 “Therefore, preventing the reduction in overall ventilation of the upper airway necessitates the evaluation of the nasopharynx, on which the effects of stenosis have not yet been clarified." (lines 74-76)

7. Authors may clarify figure 1 by improving the image quality and showing the rotations with arrows and probably the position of the nasal cavity and nasopharyngeal airway.

Reply: 

We have improved the image quality and added new explanations to Figure 1 (line 97-103).

Changes: 

Added rotations with blue arrows and the position of the nasal cavity (pink area) and nasopharyngeal airway (purple area).

8. In line 134, the atmospheric pressure mentioned is inaccurate.

Reply:

In accordance with the reviewer's comment, we have corrected from " 1.013×10-5 Pa" to "1.01325×10-5 Pa ". (in Airflow simulation section, line 152)

9. All the terms such as Pamin, Pabp, Paup, and CSA-PNA are recommended to be denoted in a more clear and readable way.

Reply: 

We have changed the terms throughout the manuscript including figures as follows:

Symbol (change from) Symbol (change to)

Nos NOS

Nval NV

PNA・PNS NP

Paup PAt

Pamin PAmin

Paup’ PAt’

Pabp PAb

ΔP nose ΔPNose

ΔP pharynx ΔPPharynx

ΔP all ΔPAll

10. All figure captions should be re-written in the form of complete sentence(s) and not phrases, where possible. Also, figure captions are better to be written as one single paragraph and not multiple paragraphs.

Reply: 

In accordance with the reviewer's comment, all figure captions have been rewritten in the form of complete sentences and/or a single paragraph. We have summarized correction of the caption in the file labeled “figure captions.docx”.

11. All parameters studied in the model can be presented in a nomenclature in the beginning or a table in the text to avoid repeating the definitions in every figure caption and at the end of each Table. For instance, alpha is defined in both lines 261 and 270 repeatedly.

Reply: 

We have added new Tables (Tables 1 and 2, please see pages 14-16), which outline definitions of landmarks and measurements variables, and removed the definitions.

12. In Table 1, units should be mentioned in front of each parameter rather than in separate cells.

Reply: 

As requested, we have removed units in separate cells and placed it in front of each parameter in Table 1 (changed to Table 3), and in Table 2 (changed to Table 4), too (Please see pages 19-20 and 22-23).

13. The quality of all figures should be improved.

Reply: 

We have corrected the quality issues pointed out in the figures.

14. Table 1 should be cross-referenced in line 243, where the data is reported.

Reply: 

We have cross-referenced Table 1 (change to Table 3) in line 251.

15. In figure 8, appropriate labels should be added to both X and Y axes instead of explaining in the figure caption.

Reply: 

We have added a label for “α”on the x-axis and “Pressure drop” on the y-axis in Figure 9 (changed from Figure 8).

16. The statement in lines 264-265 needs to be edited. Also, Figure 9 should be cross-referenced as the data is reported in lines 264-265.

Reply: 

We have revised the contents of this part to clarify and cross-referenced. 

"The closer the α-value is to 1, the smaller the pressure drop irrespective of the area (Fig. 9). " (line 262).

17. The two statements in lines 301 to 304 give the same message and hence, can be merged into one sentence.

Reply: 

In accordance with the reviewer's comment, we have merged the sentences into one sentence as follows:

“Therefore, the magnitude of ΔP was evaluated as the level of airway resistance, so the pre- and postoperative comparisons in this study revealed a postoperative improvement in the ventilation of the entire upper airway (Fig. 7).” (lines 282-284)

18. The statement in lines 299-300 seems to disagree with the statement in lines 310-312. Please clarify.

Reply: 

The airway resistance was proportional to ΔP because the flow rate was constant in this study, so we could compare the PRE and POST models. The airway resistance is inversely proportional to the airway radius based on the H–P law. The CSA of the nasopharynx (PNA, NP) is larger than that of the nasal cavity (NOS, NV). Thus, the nasopharynx has a lower impact on ΔPAll. If the stenosis is not large, the pressure drop will also be low when a is close to 1, as shown in Fig. 9 (changed from Fig. 8).

19. The statements in lines 329-331 need references.

Reply: 

We have added references [35,36] in the paragraph (line 316).

The discussion section of the manuscript now contains the following references: 

35. Kleinstreuer C, Zhang Z. Airflow and particle transport in the human respiratory system. Annu Rev Fluid Mech. 2010; 42: 301–334.

36. Cui XG, Gutheil E. Large eddy simulation of the unsteady flow-field in an idealized human mouth-throat configuration. J Biomech. 2011; 44: 2768–2774.

20. In line 337, please specify how it was found that the flow was turbulent. Please report the Re number if that is used.

Reply: 

In accordance with the reviewer's comment, we have added the explanation in the Result as follows:

“In the STENOSIS -15mm model of patient 1 and the STENOSIS -10mm model of patient 2 in Fig. 11, the Reynolds numbers of the nasopharynx region were about 11200 and 6800, respectively.” (in Results, Flow field, lines 273-274)

We appreciate all of your insightful comments. We worked hard to be responsive to them. Thank you for taking the time and energy to help us improve the paper.

Comments from Reviewer #2 

The manuscript offers good insights into the effect of changes in airway morphology due to the repositioning of bone fragments which is beneficial in predicting the effect of such operation on patients ventilation, but the number of cases used in the study is quite small which makes it quite hard to get more generalized conclusions. However, the study approach is quite promising for the future of corrective surgeries.

Reply: 

Thank you very much for your invaluable comments. We found them quite useful as we approached our revision.

1- In the abstract, page (2) line (32-33),” the rate of change in the cross-sectional area of the mass extending from the nasopharynx to oropharynx approximated 1”. It is not clear how the rate of change in the cross-sectional area was calculated?

Reply: 

We have rewritten an explanation of α in the abstract (lines 33-34).

2- In the second paragraph in the introduction, page (4) line (55-69), this paragraph is a bit confusing to the reader, as it is not clear what the authors are trying to address in this paragraph and how it is related to the current work.

Reply: 

As pointed out by the reviewer, we have rewritten the whole paragraph to clarify its meaning. The changes are as underlined as follows:

“A high incidence of mandibular skeletal prognathism (Class III) has been reported in Japan: it is treated using posterior repositioning of the mandible (single-jaw mandibular setback osteotomy). When upper and lower jaw osteotomy (two-jaw surgery) is applied because of a severe discrepancy of the jaws, the maxilla is moved upward and/or backward and the mandible is moved backward. Although there is a lack of clear evidence that corrective jaw surgery causes OSA, it is clear that posterior surgical repositioning of the mandible leads to postoperative narrowing of the upper airway. Therefore, except nasopharynx, several studies have reported the relationship between stenosis of the nasal cavity, oropharynx, or hypopharynx and OSA [3,8,15,16]. On the other hand, in the case of maxillary skeletal prognathism (Class II), the maxilla is moved upward and/or backward or the mandible is moved forward in single-jaw surgery. If two-jaw surgery is required, the maxilla is moved upward and/or backward and the mandible is moved forward. One study reported that patients with Class II have smaller pharyngeal airway volume due to the maxillofacial morphology, which is more likely to lead to OSA compared to the Class I and III skeletal relationships [17]. However, the effect of surgical repositioning of the jaw on OSA has not been elucidated. Despite maxillary impaction, anterior repositioning of the mandible in patients with a Class II skeletal relationship may improve the respiratory status during sleep by expanding the volume of the pharynx. On the other hand, in Class II, posterior and/or superior repositioning of the maxilla may lead to narrowing of the nasal cavity and nasopharynx as was observed in Class III with a reduction in the volume of the airway in the nasal cavity and the most posterior point on the posterior nasal spine (PNS) [4]. Nasal airflow and the cross-sectional area of the nasal cavity decrease when the degree of maxillary impaction exceeds a certain limit [2]. Thus, repositioning of the maxilla may reduce the volume of the entire upper airway changes with the degree of maxillary impaction and mandibular position. The nasopharynx is thought to be susceptible to the movement of both jaws due to its location. Therefore, preventing the reduction in overall ventilation of the upper airway necessitates the evaluation of the nasopharynx, which the effects of stenosis have not yet been clarified.” (in introduction, lines 54-76)

3- Figure (1) need to be more intuitive by adding a color legend to address reactionary counter-clockwise rotation during postero-superior repositioning.

Reply: 

In accordance with the Reviewer’s comment, Fig. 1 has been revised to show the movement more clearly and the associated changes in the airway. We have also added the following notes (lines 99-103):

black line, pre-surgery (before mandibular autorotation); red line, post-surgery; blue circle, center of mandibular autorotation; blue arrow, direction of autorotation; green line, after mandibular autorotation; gray line; post-surgery in the figure of airway changes ; pink area, preoperative nasal cavity; pink hatched area, postoperative nasal cavity； pink arrow, direction of nasal cavity change; purple area, preoperative nasopharyngeal airway; purple hatched area, postoperative nasopharyngeal airway; purple arrow, direction of pharyngeal change

4- typo error line (90) page (6),” Surgical impaction of the maxilla and the reaction of the mandible ae illustrated schematically”.

Reply: 

This error has been corrected in accordance with the reviewer's comment from "ae" to "are." (in introduction, line 98)

5- It has to be mentioned clearly that the study was performed on females only because the authors mentioned that the study was performed on 3 females.( page 7, line 100).

Reply: 

Accordingly, we have added “females” to abstract section (line 24). As well, we have specified “three patients, all female” in the participants section (line 112).

6- Figure illustrating the mesh is needed to provide information regarding this information “Three layers of the tetrahedral/hybrid tetrahedral-prism”, mentioned in line 129, page(8).

Reply: 

In accordance with the reviewer's comment, we have included a new Figure 3 to illustrate the mesh (lines 146-148 and Figure 3).

7- Information about the element size and mesh independence study need to be added.

Reply: 

We have changed the following text to the Three-dimensional models section (lines 141-144) from: 

“Three layers of the tetrahedral/hybrid tetrahedral-prism meshes were generated so that even the area near the wall possessed sufficient resolution.”

to

“The volume mesh of the airway had around 7400000 elements. The unstructured tetrahedral/prism hybrid mesh of the airway model was generated. Three layers of the prism mesh was placed near the wall so that even the area near the wall possessed sufficient resolution (Fig. 3). The cell size of the prism region was adjusted to attain a dimensionless wall distance (y+) value less than 1.”

8- Reason for choosing these assumptions is steady and need to be clarified in the text. Line (135), page (9).

Reply: 

Lee et al explained that significant change was not observed in flow pattern distribution between steady and unsteady calculation at inhalation phase. Therefore, we have chosen steady flow to simplify calculations. We have also added the following underlined text and reference (lines 152-156):

“The following physical properties were set in the model: steady flow of an incompressible Newtonian fluid with a density of 1.205 kg/m3 and viscosity of 1.822 × 10-5 Pa·s based on a previous study [26]. Lee et al. [26] explained that significant change was not observed in flow pattern distribution between steady and unsteady calculation at the inhalation phase.”

26. Lee JH, Na Y, Kim SK, Chung SK. Unsteady flow characteristic through a human nasal cavity. Respiratory Physiology & Neurobiology. 2010;172: 136-146.

9- In page (10) line (162-163), the Outlet boundary condition is used a free outlet and p=0 in one of the cases, can you clarify the reason behind using different outlet boundary conditions.

Reply: 

Thank you for this comment. We have also added the following underlined text and reference (lines 184-185):

“Because back flow occurred at the outflow boundary, the pressure boundary condition (P=0) was adapted in the high-stenosis model based on reality.”

10- Figure (6), a y-axis label need to be added and the x-axis.

Reply: 

We have added a label for “Model”on the x-axis. Also, “Pressure drop” and “CSA” have been added on the y-axis in Figure 7 (changed from Figure 6).

11- Can you indicate how the STENOSIS -1mm, -3mm is measured in the figure (3), it seems to be a distance, so can you clarify how this is measured

Reply: 

We have added the following text to clarify how to create the STENOSIS models:

“As shown in Fig.4, the nasopharynx of the STENOSIS model (indicated by the rectangle in the inset) was trimmed mainly around the PNS by the amount indicated by the red asterisks (where each asterisk equals to the amount of trimming for each model, e.g., in STENOSIS -1 mm model, the asterisk means narrowing the thickness by 1 mm).” (lines 193-196)

“With a focus on the nasopharynx region of the STENOSIS model, the area (surrounded by the orange dotted circle) was trimmed by the length of the asterisk around the posterior nasal spine (PNS) in the sagittal plane. The red asterisk indicates the amount of trimming for the nasopharynx of the STENOSIS model.” (lines 198-200)

12- In page (25), line (327), the assumption of rigid upper airways is quite obsolete; as some studies have investigated the effect of upper airways tissue motion (The effects of upper airway tissue motion on airflow dynamics). Also, most of the experimental studies use flexible materials for manufacturing upper airway replicas’.

Reply: 

The use of a rigid model may be a limitation of the present study. The influence of model conditions will be a consideration for our future study. We have therefore added the following text and references (in discussion, lines 307-311):

“Note that in this study, we have assumed a rigid wall and steady state. The mechanical properties of the pharynx wall are difficult to determinate, because it is regulated by a complex interplay between whether enclosed in a bony structure, wall thickness, airspace cross-sectional areas, and tissue pressure [32,33]. Therefore, the compliance effects have not been considered and have been simplified. This behavior is an important aspect that should be taken in consideration in future studies.”

32. White DP, MK Younes. Obstructive sleep apnea. Compr Physiol. 2012;2: 2541–2594.

33. Woodson BT. A method to describe the pharyngeal airway. Laryngoscope 2015;125: 1233–1238.

13- Body mass index (BMI) for the cases tested in this study needs to be mentioned.

Reply: 

We have supplemented the Materials and Methods section (line 115) and Table 3 with data of BMI.

Thank you once again for your valuable comments and suggestions.

---

## [Decision Letter · Decision Letter 1]

8 Apr 2022

PONE-D-21-36984R1Computational fluid dynamic analysis of the nasal respiratory function before and after postero-superior repositioning of the maxillaPLOS ONE

Dear Dr. Oshima,

Thank you for submitting your manuscript to PLOS ONE. After careful consideration, we feel that it has merit but does not fully meet PLOS ONE’s publication criteria as it currently stands. Therefore, we invite you to submit a revised version of the manuscript that addresses the points raised during the review process.

The error is minor, and one reviewer has noted you have had the wrong unit for the atmospheric pressure. Please kindly make the amendment, or clarify the text. ==============================

We look forward to receiving your revised manuscript.

Kind regards,

Shaokoon Cheng

Academic Editor

PLOS ONE

Journal Requirements:

Reviewers' comments:

Reviewer's Responses to Questions

**Comments to the Author**

1. If the authors have adequately addressed your comments raised in a previous round of review and you feel that this manuscript is now acceptable for publication, you may indicate that here to bypass the “Comments to the Author” section, enter your conflict of interest statement in the “Confidential to Editor” section, and submit your "Accept" recommendation.

Reviewer #1: (No Response)

Reviewer #2: All comments have been addressed

2. Is the manuscript technically sound, and do the data support the conclusions?

Reviewer #1: Yes

Reviewer #2: Yes

3. Has the statistical analysis been performed appropriately and rigorously? 

Reviewer #1: N/A

Reviewer #2: N/A

4. Have the authors made all data underlying the findings in their manuscript fully available?

Reviewer #1: Yes

Reviewer #2: Yes

5. Is the manuscript presented in an intelligible fashion and written in standard English?

Reviewer #1: Yes

Reviewer #2: Yes

6. Review Comments to the Author

Reviewer #1: The manuscript is significantly improved.

However, the atmospheric pressure mentioned in page 10 line 152 is still inaccurate and should be changed to 1.013 x 10^+5 Pa.

Reviewer #2: (No Response)

7. PLOS authors have the option to publish the peer review history of their article (what does this mean?). If published, this will include your full peer review and any attached files.

Reviewer #1: No

Reviewer #2: **Yes: **khalid Elserfy

---

## [Author Response · Author response to Decision Letter 1]

11 Apr 2022

Thank you for giving me the opportunity to submit a revised draft of my manuscript titled “Computational fluid dynamic analysis of the nasal respiratory function before and after postero-superior repositioning of the maxilla” to PLOS ONE. 

We appreciate the time and effort that you and the reviewers have dedicated to providing your valuable feedback on our manuscript. We have revised the manuscript according to the comments from editors and reviewers. 

We have highlighted the changes within the manuscript. 

Here is a point-by-point response to the journal requirements and the reviewers’ comments. 

Journal Requirements 

Reply: 

We have ensured that our reference list is complete and correct. Additional reference lists have noted as follows:

20. Hoppenreijs TJ, Stoelinga PJ, Grace KL, Robben CM. Long-term evaluation of patients with progressive condylar resorption following orthognathic surgery. Int J Oral Maxillofac Surg. 1999;28: 411-418. doi: 10.1034/j.1399-0020.1999.280602.x.

Comments from Reviewer #1 

The manuscript is significantly improved.

However, the atmospheric pressure mentioned in page 10 line 152 is still inaccurate and should be changed to 1.013 x 10^+5 Pa.

Reply: 

Thank you very much for your kind suggestion.

In accordance with the reviewer's comment, we have corrected from " 1.013×10-5 Pa" to "1.013×105 Pa ". (in Airflow simulation section, page 10 line 152)

Comments from Reviewer #2 

(No Response)

Reply: 

Thank you very much for reviewing our manuscript.

We look forward to hearing from you in due time regarding our submission and to respond to any further questions and comments you may have. 

Sincerely, 

Marie Oshima

---

## [Editor Report · Decision Letter 2]

13 Apr 2022

Computational fluid dynamic analysis of the nasal respiratory function before and after postero-superior repositioning of the maxilla

PONE-D-21-36984R2

Dear Dr. Oshima,

We’re pleased to inform you that your manuscript has been judged scientifically suitable for publication and will be formally accepted for publication once it meets all outstanding technical requirements.

Kind regards,

Shaokoon Cheng

Academic Editor

PLOS ONE

---

## [Editor Report · Acceptance letter]

18 Apr 2022

PONE-D-21-36984R2 

Computational fluid dynamic analysis of the nasal respiratory function before and after postero-superior repositioning of the maxilla 

Dear Dr. Oshima:

I'm pleased to inform you that your manuscript has been deemed suitable for publication in PLOS ONE. Congratulations! Your manuscript is now with our production department. 

Kind regards, 

on behalf of

Dr. Shaokoon Cheng 

Academic Editor

PLOS ONE